# Role of NS1 antibodies in the pathogenesis of acute secondary dengue infection

Deshni Jayathilaka[1], Laksiri Gomes[1], Chandima Jeewandara[1], Geethal.S.Bandara Jayarathna[1], Dhanushka Herath[1], Pathum Asela Perera[1], Samitha Fernando[1], Ananda Wijewickrama[2], Clare S. Hardman[3], Graham S. Ogg[3] & Gathsaurie Neelika Malavige [1,3]

The role of NS1-specific antibodies in the pathogenesis of dengue virus infection is poorly understood. Here we investigate the immunoglobulin responses of patients with dengue fever (DF) and dengue hemorrhagic fever (DHF) to NS1. Antibody responses to recombinant-NS1 are assessed in serum samples throughout illness of patients with acute secondary DENV1 and DENV2 infection by ELISA. NS1 antibody titres are significantly higher in patients with DHF compared to those with DF for both serotypes, during the critical phase of illness. Furthermore, during both acute secondary DENV1 and DENV2 infection, the antibody repertoire of DF and DHF patients is directed towards distinct regions of the NS1 protein. In addition, healthy individuals, with past non-severe dengue infection have a similar antibody repertoire as those with mild acute infection (DF). Therefore, antibodies that target specific NS1 epitopes could predict disease severity and be of potential benefit in aiding vaccine and treatment design.

[1] Centre for Dengue Research, University of Sri Jayewardenepura, Nugegoda 10100, Sri Lanka. [2] National Institute of Infectious Diseases, Angoda 10250, Sri Lanka. [3] MRC Human Immunology Unit, Weatherall Institute of Molecular Medicine, Oxford NIHR Biomedical Research Centre, Oxford OX3 9DS, UK. These authors contributed equally: Deshni Jayathilaka, Laksiri Gomes. The authors jointly supervised this work: Graham S. Ogg, Gathsaurie Neelika Malavige. Correspondence and requests for materials should be addressed to G.N.M. (email: gathsaurie.malavige@ndm.ox.ac.uk)

Dengue virus (DENV) is one of the most rapidly emerging viral infections worldwide, infecting 390 million individuals annually. Despite such obvious need, there is currently no licensed specific drug for the treatment of this potentially fatal disease[1]. A tetravalent live attenuated yellow fever and dengue chimeric vaccine has been recently licensed in some countries, however, it has poor efficacy in naive individuals and efficacy depends on DENV serotype[2]. In addition, the vaccine manufacturer recently suggested that this vaccine should be avoided in dengue naive individuals, due to the likelihood of disease enhancement[3]. One of the major challenges faced in the development of an efficacious dengue vaccine is our poor understanding of what constitutes a protective immune response.

DENV infections result in a spectrum of disease ranging from subclinical inapparent presentation, through mild dengue fever (DF) to severe dengue hemorrhagic fever (DHF), which is characterized by an increase in vascular permeability resulting in shock and organ dysfunction[4]. Understanding the molecular pathway that leads to development of vascular leak would provide a major step forward in the development of effective dengue treatments. One such potential target is dengue non-structural protein 1 (NS1), a secreted glycoprotein used as a diagnostic marker of dengue infection, appearing early in the serum before antibodies are generated[5]. NS1, synthesized as a monomer, forms a dimer in the ER lumen and hexamer in the serum. NS1 is thought to function as a cofactor for viral RNA replication[6]. NS1 has been shown to trigger cytokine release and contributes directly to vascular leak through binding TLR4 and engaging the endothelial glycocalyx[7,8]. In addition, in vitro data show that some antibodies against NS1 cross-react with endothelial cells and induce apoptosis, which is suggested to contribute to endothelial dysfunction and vascular leak[9]. Interestingly, NS1 is important for viral evasion of complement activation by binding mannose-binding lectin and preventing neutralization of DENV via the lectin pathway[10]. However, some studies have shown that NS1 activates complement, a process that is further enhanced by NS1-specific antibodies[11].

Secondary dengue infections incur an increased risk of developing DHF, which supports the hypothesis that NS1-specific antibodies derived from primary infection may, upon expansion following secondary challenge, play a role in disease pathogenesis[12,13]. However, apoptosis of the endothelium and deposition of immune complexes in the endothelium have not been demonstrated in autopsy studies of fatal dengue[14]. In contrast to the data in the above studies, some investigations in dengue mouse models have shown that mice injected with sera from dengue-infected mice or monoclonal antibodies to NS1, have significantly reduced vascular leak, implying a protective role for antibodies to NS1[15]. It is currently not clear if antibodies targeted to specific regions of dengue NS1 protein may play differential roles in the protection or pathogenesis of dengue fever in humans.

The majority of human DENV-specific antibodies are directed against DENV envelope and PrM proteins, and these responses have been relatively well-characterized[16–19]. However, NS1-specific antibodies have not been thoroughly investigated despite constituting 27% of antibody responses[20]. There are limited data on the target, isotype and function of anti-NS1 antibodies in acute secondary dengue infections, which account for most cases of severe dengue infection.

Interestingly, Dengvaxia®, the only registered dengue vaccine currently available, does not generate DENV–NS1-specific antibodies[21], and some have speculated that the failure to raise NS1-specific antibodies may contribute to the lower-than-expected efficacy of this vaccine[22]. In order to address these questions, here we investigate NS1-specific antibody responses in a cohort of patients with acute secondary dengue (DENV1 and DENV2) infection and study how NS1-specific antibody responses evolve during the course of illness in relation to clinical disease severity. We further characterize anti-NS1 antibody responses in patients and healthy individuals with varying severity of past dengue infection to identify epitope regions within the NS1 protein that associate with disease progression or protection.

## Results

**Dengue patient cohort**. Prior to investigating the NS1 antibody responses in a patient cohort, we evaluated the NS1 antibody levels in a healthy population to determine NS1 antibody levels at baseline. The NS1 antibody levels were assessed by an in-house ELISA, in DENV-seronegative individuals ($n = 20$), in those who were seropositive for DENV but had never been hospitalized for a febrile infection and were therefore considered to have past non-severe dengue (NSD) ($n = 36$) and those who had a past history of DHF and were considered to have past severe dengue (SD) ($n = 34$). The median NS1 antibody titres of seronegative individuals was 0.37 (IQR: 0.21 to 0.4 OD at 405 nm). Healthy individuals with past SD (median 1.6, IQR: 0.78–2.2 OD at 405 nm) had significantly higher antibody titres ($p = 0.004$ using the Mann–Whitney U test (two tailed)) compared with healthy individuals with past NSD (median 0.91, IQR: 0.65–1.36 OD at 405 nm) (Fig. 1a).

To investigate NS1 antibody responses in patients with acute secondary dengue infection, we recruited 76 patients with acute dengue, and through analysis of patient IgM:IgG antibody ratios determined that 55/76 patients had a secondary dengue infection (IgM:IgG antibody ratio < 1.2)[23] and were included in the analyses of NS1 antibody responses. Twenty patients of the secondary cohort had acute DENV1 infection, and 35 had acute DENV2 infection. Ten patients with acute DENV1 infection had DHF, while 10 had DF and 23 patients with an acute DENV2 infection had DHF and 12 had DF (Table 1). Although differentiating primary and secondary dengue based on IgM and IgG ratios may not be optimum and may not be 100% accurate in some patients, currently, there are no T-cell-based assays for such purpose. Although we had previously developed a T-cell-based assay to determine past infecting serotype[24], we found that this assay was not suitable to be used in patients with acute infection, due to absent or low T-cell responses related to lymphopenia[25,26].

**Kinetics of NS1 antibodies in acute secondary dengue infection**. Serum samples were collected daily from the patients within our cohort, and levels of DENV1 (Fig. 1b) or DENV2 (Fig. 1c) NS1-specific IgG antibodies were measured. We found that NS1 antibody responses in both acute secondary DENV1 and DENV2 infection increased profoundly during the critical phase and were significantly higher in DHF patients compared with those with DF (Fig. 1b, c). In DENV1-infected DHF patients, the serum concentration of NS1 antibodies started to rise 5.5 days after the onset of fever. NS1 antibody levels were significantly higher in patients with DHF than in those with DF on day 6 ($p = 0.003$) and 7 ($p = 0.002$) of illness, when analysed using Holm–Sidak methods which correct the $p$-value in multiple comparisons ($p = 0.05$), which represents the critical phase (Fig. 1b). The same pattern was observed in patients with an acute secondary DENV-2 infection, although the difference in antibody titres between patients with DHF and DF was less significant than in DENV1 infection (day 6, $p = 0.04$). In this cohort, it was shown that onset of the critical phase in patients with DENV-2 infection occurs significantly earlier than in acute DENV1 infection[27]. Accordingly, here we find that NS1 antibody titres rise a day earlier in

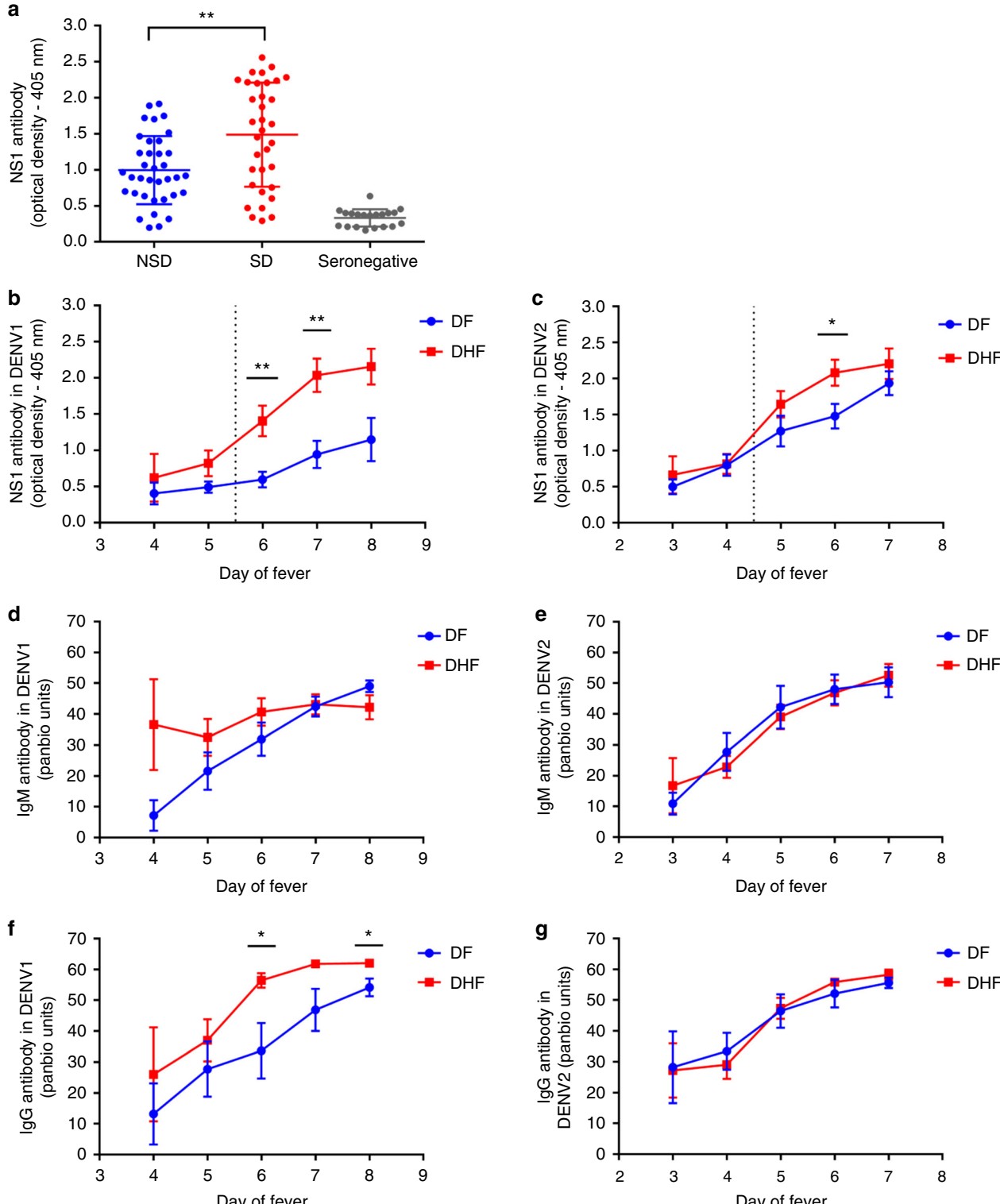

**Fig. 1** Kinetics of DENV NS1-specific antibody responses in acute dengue. **a** NS1 antibody levels in patients with past severe dengue (SD) ($n = 34$), non-severe dengue (NSD) ($n = 36$) and dengue seronegatives ($n = 20$). **b** DENV1 NS1-specific IgG antibody levels, measured in patients with an acute secondary DENV1 infection causing DF ($n = 10$) or DHF ($n = 10$) using ELISA. **c** DENV2 NS1-specific IgG antibody levels were measured in patients with an acute secondary DENV2 infection causing DF ($n = 12$) or DHF ($n = 23$) using ELISA. DENV-specific IgM antibody (Panbio unit) in patients with an acute secondary DENV1 (**d**) and DENV2 (**e**). DENV-specific IgG antibody levels were measured in patients with an acute DENV1 (**f**) and DENV2 (**g**) infection. Differences between serial values of NS1-specific antibodies and IgM and IgG in patients with DHF and DF were compared using the Holm–Sidak method. The vertical dotted line represents the day in which the patients entered the critical phase. NS1 antibody levels in patients with DHF are indicated in red, and in those with DF in blue. Error bars indicate mean and standard error of mean (SEM). *$P < 0.05$, **$P < 0.01$

**Table 1 Clinical and laboratory characteristics of secondary dengue patients with DHF and DF recruited for the study**

| Clinical findings | DHF (n = 33) | DF (n = 22) |
|---|---|---|
| Vomiting | 14 (42.42%) | 5 (22.72%) |
| Abdominal pain | 21 (63.63%) | 5 (22.72%) |
| Hepatomegaly | 28 (84.84%) | 0 |
| Bleeding manifestations | 1 (03.03%) | 3 (13.63%) |
| Pleural effusion | 18 (54.54%) | 0 |
| Ascites | 33 (100.0%) | 0 |
| *Lowest platelet count* | | |
| <20,000 cells/mm³ | 23 (69.69%) | 2 (9.09%) |
| 20,000–50,000 | 10 (30.30%) | 9 (40.90%) |
| 50,000–100,000 | 0 | 8 (36.36%) |
| >100,000 | 0 | 3 (13.63%) |
| *Lowest lymphocyte count* | | |
| < 750 | 21 (63.63%) | 9 (40.90%) |
| 750–1500 | 10 (30.30%) | 13 (59.09%) |
| >1500 | 2 (6.06%) | 0 |
| *Infecting serotype* | | |
| DENV1 | 10 (30.30%) | 10 (45.45%) |
| DENV2 | 23 (69.69%) | 12 (54.54%) |

DHF patients with DENV-2 (day 4.5) than DENV1 infection (Fig. 1b, c).

We also semi-quantitatively measured DENV-specific IgM and IgG antibody levels throughout the course of illness, using the Panbio IgM and IgG capture ELISA. Both Panbio IgM and IgG ELISAs use DENV envelope protein as the coating antigen, and antibody titres are expressed as Panbio units. There was no significant difference in DENV-specific IgM antibody titres throughout the course of illness between patients with DHF or DF with either acute secondary DENV1 or DENV2 infection (Fig. 1d, e, respectively). Interestingly, in patients with an acute secondary DENV1 infection, IgG antibody titres were significantly higher during the critical phase on day 6 ($p = 0.019$) and day 8 ($p = 0.018$) when analysed using the Holm–Sidak method. (Fig. 1f). However, there was no difference in the DENV-specific IgG antibody titres in patients with an acute secondary DENV2 infection (Fig. 1g).

In order to measure the association of NS1-specific IgG, and DENV-specific IgM and IgG antibodies, we also correlated the NS1 antigen levels, measured by Panbio NS1 capture ELISA and viral loads in these patients throughout the course of illness as previously described[28]. We found that NS1 antigen levels were higher in patients with DF compared with DHF in acute DENV1 (Supplementary Fig. 1a), whereas the NS1 antigen levels were higher in patients with DHF compared with those with DF in acute DENV2 infection (Supplementary Fig. 1b). Although viral loads in DENV1 infection showed a similar pattern to NS1 antigen (Supplementary Fig. 1c), this was not seen in acute DENV2 infection, where viral loads in patients with DF and DHF were similar (Supplementary Fig. 1d). As expected, there was a significant inverse correlation of NS1 antibody levels with NS1 antigen levels (Spearman's r = −0.45, $p = 0.005$) (Supplementary Fig. 1e), viral loads (Spearman's r = −0.79, $p < 0.0001$) in patients with acute DENV1 infection who had DF, and also with NS1 antigen levels (Spearman's r = −0.60, $p < 0.0001$) (Supplementary Fig. 1f) and viral loads (Spearman's r = −0.65, $p = 0.0003$), in patients with acute DENV1 infection resulting in DHF. The same association was seen in NS1 antigen levels (Spearman's r = −0.50, $p = 0.0006$) (Supplementary Fig. 1g), viral loads (Spearman's r = −0.58, $p < 0.0001$) in patients with acute DENV2 infection who had DF, and also with NS1 antigen levels (Spearman's r = −0.80, $p < 0.0001$) (Supplementary Fig. 1h) and viral loads (Spearman's r = −0.72, $p < 0.0001$), in patients with acute DENV2 infection resulting in DHF.

NS1 antibody titres of DHF and DF patients with acute secondary DENV2 infection inversely correlated with platelet counts (Spearman's r = −0.3, $p = 0.008$ and Spearman's r = −0.4, $p = 0.009$, respectively) (Fig. 2a, b). In acute secondary DENV1 infection, NS1 antibody titres inversely correlated with the platelet counts in patients with DHF (Spearman's r = −0.3, $p = 0.03$), but there was no significant correlation in patients with DF (Spearman's r = −0.17, $p = 0.30$) (Fig. 2c, d).

**Characterization of antibody responses to NS1 peptides.** Our observation that NS1 antibody responses were significantly increased in patients with acute secondary DENV1 and DENV2 infection during the critical phase of illness, leads us to investigate the epitopes recognized by anti-NS1 antibodies. The NS1 antibody levels were measured at day 4 (DENV1) and day 3 (DENV2), as it was felt important to measure the NS1 epitopes in patients with DF or DHF before they proceeded to the critical phase (while all patients had DF and had not progressed to vascular leakage). They were again measured on day 7 in patients with DENV1 and day 6 in patients with DENV2, as this was when the most significant difference in NS1 antibody titres was observed between DF and DHF patients. We performed ELISA on patient serum samples to assess antibody recognition of a series of overlapping peptides constituting the entire length of the NS1 protein sequence.

We found that in both acute secondary DENV1 and DENV2 infection, there were significantly higher antibody titres in DF patients (Figs. 3, 4, 5 and 6) recognizing certain regions, or peptides, of NS1 compared with DHF patients. The epitope regions, for which a significant difference in antibody responses between DF and DHF patients was detected, were similar for DENV1 and DENV2 overlapping NS1 peptides (Figs. 3, 4, 5 and 6). During the febrile phase, patients with DF had significantly higher responses to DENV1 NS1 peptides spanning aa 297–311 (pep 53), aa 318–334 (pep 57) and aa 341–353 (pep 61) (Fig. 3g, h). In acute DENV2 infection during the febrile phase, patients with DF had significantly higher responses to aa 62–102 (pep 9–12), aa 177–210 (pep 25–27), aa 236–280 (pep 33–37) and 289–306 (pep 41) (Fig. 4b, d, e, f). In comparison, when comparing the NS1 antibody responses in the critical phase of illness, patients with DF had significantly higher responses to DENV1 NS1 peptides spanning aa 73–107 (pep14, 15 and 17) and aa 115–137 (pep 21–22) regions and DENV2 NS1 peptides aa 33–111 (pep 5–13), aa 115–141 (pep 16 and pep 18) and aa 146–163 (pep 21) (Figs. 5b, 5c and 6a–c). In addition, patients with DF had significantly higher antibody titres to the epitope region represented by aa 152–180 of both DENV1 and DENV2 NS1 (peptide 26–31 in DENV1 NS1 and peptides 21–24 in DENV2 NS1) (Figs. 5d and 6c).

In the febrile phase in DENV1 infection, patients with DF had significantly higher antibody responses to the distal C-terminal region (aa 277–324), whereas in DENV2 infection, patients with DF also had significantly higher responses to the C-terminal region (aa 236–280), but also to regions in the N-terminal region (aa 62–102). In the critical phase, notably patients with DF, as a result of either DENV1 and DENV2 infection, had significantly higher antibody titres to the distal C-terminal region (aa 230–300) represented by pep 41–48 of DENV1 and pep 33–42 and pep 44–45 in DENV2 (Figs. 5f, 6e, f). In contrast, no differences were observed between patients with DF and DHF with DENV1 or DENV2 to the proximal end of C-terminal NS1 during the febrile or the critical phase (Figs. 5e and 6d).

Although there were similarities in recognition of NS1 antigen epitopes in patients with acute DENV1 and acute DENV2, there were also notable differences. In order to investigate if these

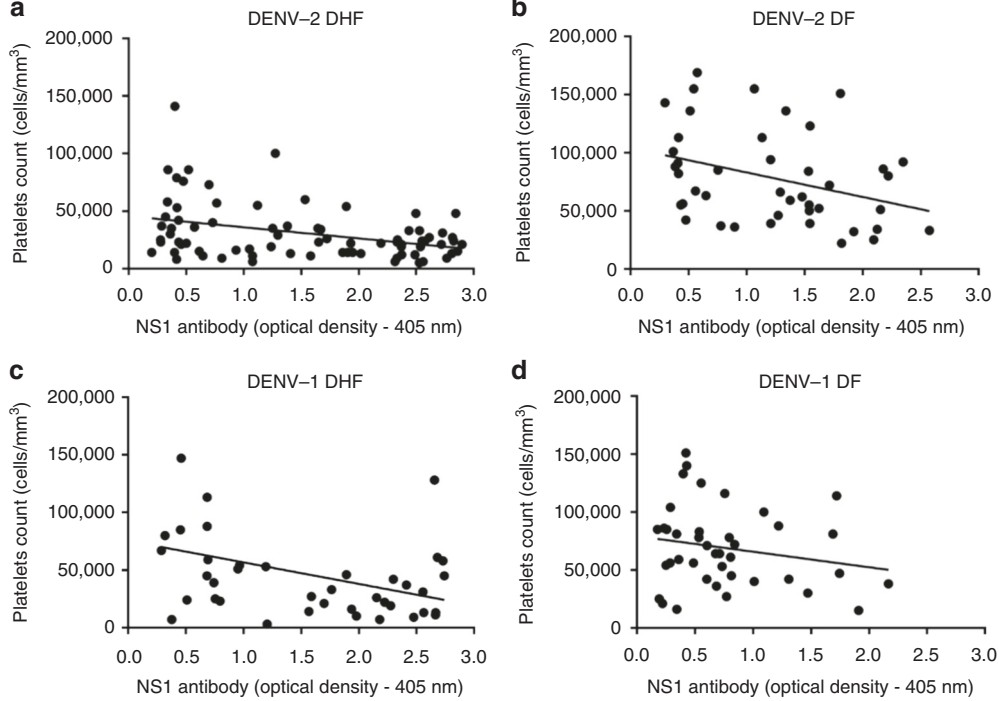

**Fig. 2** Correlation of NS1 antibody titres with platelet counts. Correlation of NS1 antibody titres in patients with acute secondary DENV1 and DENV2 infection with platelet counts. Serum anti-NS1 antibody titres were measured throughout the course of illness by ELISA and correlated with circulating platelet counts in patients with (**a**) DHF due to DENV2, $n = 23$ (Spearmans' $r = -0.3$, $p = 0.008$), (**b**) DF due to DENV2, $n = 12$ (Spearman's $r = -0.39$, $p = 0.009$), (**c**) DHF due to DENV1, $n = 10$ (Spearman's $r = -0.3$, $p = 0.03$), and (**d**) DF due to DENV1, $n = 10$ (Spearman's $r = -0.17$, $p = 0.30$)

differences were due to the differences in the NS1 sequences of DENV1 and DENV2, we did multiple alignment of DENV1 and DENV2 sequences isolated in Sri Lanka during the last few years using clustal omega[29] (Supplementary Fig. 2a). As the DENV2 sequence responsible for the current epidemic has not been published, we only aligned DENV2 sequences of past DENV2 epidemics. During the febrile phase, patients with DF due to either DENV1 or DENV2 had significantly higher antibody responses than patients with DHF, directed to highly conserved regions of the NS1 (aa 297–311, aa 318–324 and aa 345–353 for DENV1 and aa 61–102, aa 170–210 and aa 236–280 in DENV2). The DENV1 and DENV2 overlapping peptides that were used to test for NS1-specific antibodies were also similar to the recent circulating DENV1 and DENV2 NS1 antigen sequences in Sri Lanka (Supplementary Fig. 2b and 2c).

**NS1 antibody responses to NS1 peptides throughout the course of illness.** Having observed that patients with acute secondary DF, compared with DHF, generated more antibodies targeted towards similar regions of DENV1 and DENV2 sequences during the critical phase of illness, we investigated how the antibody responses to these regions evolve through the course of illness. Therefore, we chose seven regions of NS1 in DENV1, represented by a combination of two or four peptides to detect antibody specificity, for which patients with DF made significantly higher antibody responses, to see the kinetics of the antibody responses. In both DF and DHF patients, responses to pool 2 (aa 121–142) followed by pool 7 (aa 341–353) elicited the highest antibody titres (Fig. 7). By using the Holm–Sidak method for analysis, we found that patients with DF had higher responses to peptide pools 2 ($p = 0.02$) and pool 7 ($p = 0.03$), representing the C-terminal end of NS1, towards the end of the critical phase than

DHF sera, suggesting that antibody responses to these immunodominant regions of NS1 could associate with protection.

**Antibody responses to recombinant and denatured NS1.** Although we found that antibodies to recombinant NS1 were significantly higher in patients with DHF during the critical phase, further analysis of NS1 epitopes showed that patients with DF had significantly higher antibody titres to certain peptides/regions of NS1. In order to determine whether the NS1 antibodies bind to conformational epitopes not represented in the peptide-binding assay, we coated 96-well plates with recombinant or denatured DENV NS1 and measured the antibody titres in the sera of patients with DF and DHF during acute secondary DENV infection. The recombinant NS1 was denatured by heating the protein at 95 °C for 15 min, as previously described[30]. The sera used for these experiments were sampled on day 7 of illness from patients with DENV1 and day 6 from patients with DENV2. The detection of antibody titres significantly decreased when the DENV1 NS1 protein capture antigen was denatured in patients with acute DENV1 infection resulting in DHF ($p < 0.0001$, using the Mann–Whitney U test) (two tailed) and DF ($p = 0.008$) (Fig. 8a). Similarly, a significant reduction in the antibody titres was seen when DENV2 NS1 protein was denatured in patients with DHF ($p = 0.0004$) although not significant in patients with DF (Fig. 8b).

**DENV NS1 IgG subclass types in patients with acute dengue.** Different IgG subclasses that form immune complexes, vary in their ability to activate complement and have different affinities for IgG receptors[31]. While IgG binding to FcγRI, FcγRIIA/C and FcγRIIIB results in activation of macrophages, monocytes and dendritic cells, IgG binding to FcγRIIB may inhibit immune

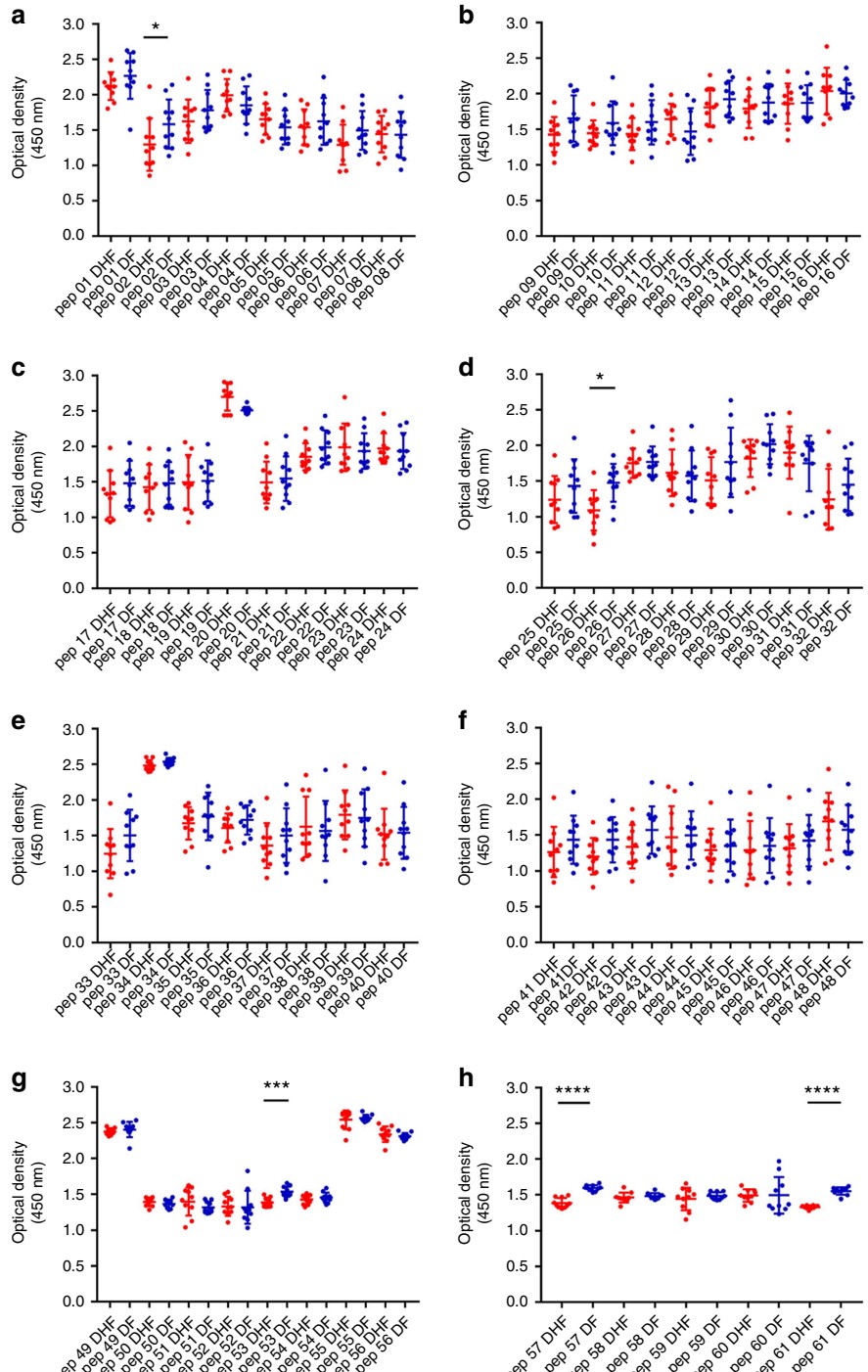

**Fig. 3** DENV1 NS1 antibody responses to NS1 peptides in the febrile phase. **a–h** DENV1 NS1-specific IgG antibody responses to 61 DENV1 NS1 overlapping peptides in patients with DF ($n = 10$) and DHF ($n = 10$) were measured by ELISA on day 4 of illness. Differences in mean values between antibody responses to overlapping peptides in patients with DF and DHF were compared using the Mann–Whitney U test (two tailed). NS1 antibody levels in patients with DHF are indicated in red, and in those with DF in blue. Error bars indicate mean and standard deviation (SD). *$P < 0.05$, **$P < 0.01$, ***$P < 0.001$, ****$P < 0.0001$

responses[32,33]. IgG3 has shown to be the predominant subclass that activated the inhibitory FcγRIIB receptors, whereas IgG1 preferentially activated the activating receptors[34]. Therefore, we investigated the NS1-specific IgG subclasses in patients with acute secondary DENV2 infection. We found that IgG1 was the predominant subclass of NS1 antibody present and NS1-specific IgG1

was significantly higher in patients with DHF ($p = 0.037$), when compared with patients with DF (Fig. 8c) by Mann–Whitney U test (two tailed). In contrast, the NS1 IgG3 subclass levels showed a trend to be higher in patients with DF (median 0.04, IQR: 0.03–0.06) compared with those with DHF (median 0.02, IQR: 0.01–0.03) although this was not significant ($p = 0.1$).

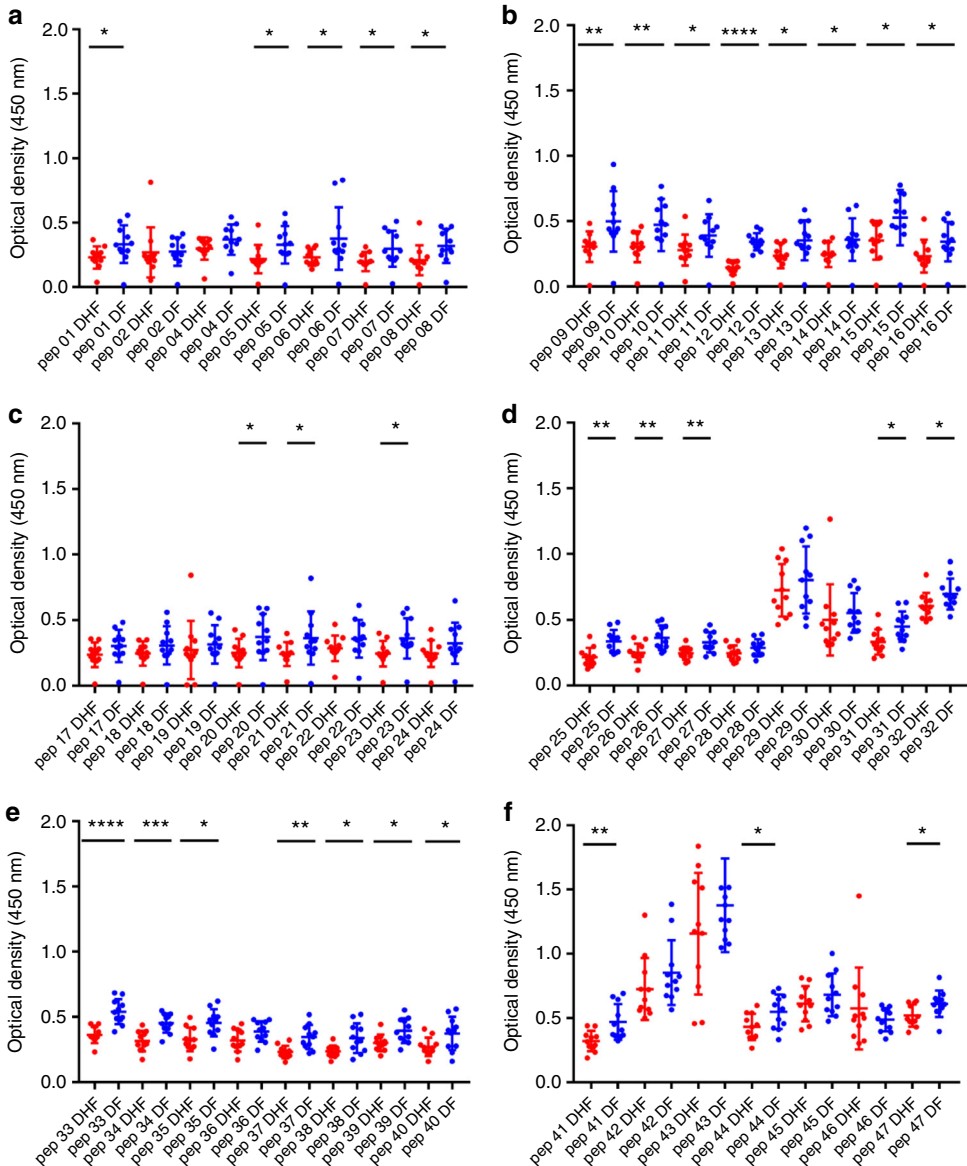

**Fig. 4** DENV2 NS1 antibody responses to NS1 peptides in the febrile phase. **a–f** DENV2 NS1-specific IgG antibody responses to 47 DENV2 NS1 overlapping peptides were measured by ELISA in patients with DF ($n = 10$) and DHF ($n = 10$) on day 3 of illness. Differences in mean values between antibody responses to overlapping peptides in patients with DF and DHF were compared using the Mann–Whitney U test (two tailed). Error bars indicate the means and standard deviation (SD). *$P < 0.05$, **$P < 0.01$, ***$P < 0.001$, ****$P < 0.0001$

**Antibody responses to NS1 in healthy individuals past dengue.**
Recurrent dengue infection is thought to be a risk factor for developing severe dengue. As mentioned above, we investigated the total NS1 antibody levels in healthy individuals who were DENV seronegative, and those with past NSD and SD. We found that those with past SD had significantly higher NS1 antibody levels than in individuals with past NSD (Fig. 1a). We then characterized the antibody responses to NS1 in 24 individuals with past NSD and 17 individuals with past SD. We cannot know when the NSD cohort contracted DENV or whether the infection was primary or secondary, due to their lack of clinical presentation.

As observed in patients with acute dengue infection (Figs. 3–6), we found that healthy participants with past SD and NSD recognized distinct regions of DENV NS1. We observed negligible levels of antibodies to the NS1 overlapping peptides in DENV-seronegative individuals (Fig. 9a–o). Patients with past SD, had

significantly higher antibody responses to regions represented by peptides 1–4, 9–16, 22–26 and 29–31. In contrast, as observed in patients with DF, those who had past NSD had significantly higher antibody responses to the distal C-terminus of NS1 compared to those with past SD, represented by peptide 33–37, 40–43, 45–48 and 58–60 (Fig. 9a–o). Therefore, while those with past SD had significantly higher antibody responses to the N-terminus of the NS1 antigen, those with past NSD had significantly higher antibodies to the peptides representing the C-terminus (Supplementary Fig. 3).

**Cross-reactivity of DENV NS1 epitopes with other flaviviruses.**
Among the non-structural proteins of the flaviviruses, NS1 has been shown to be the most highly conserved protein[35]. In order to find out if the NS1 antibody epitopes defined in this study were directed at conserved or non-conserved regions, we carried out multiple alignment of the NS1 antigen of Japanese Encephalitis

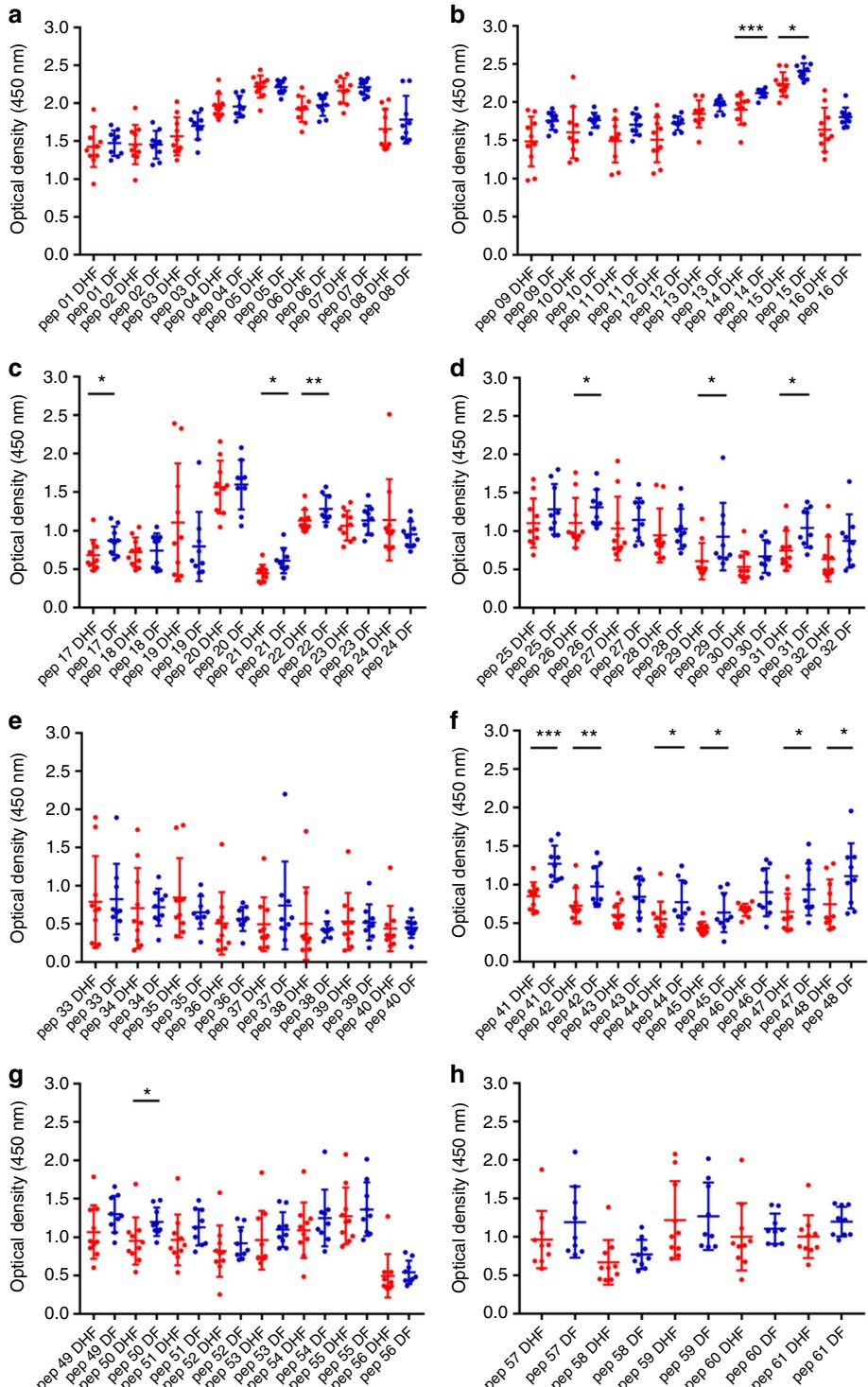

**Fig. 5** DENV1 NS1 antibody responses to NS1 peptides in the critical phase. **a–h** DENV1 NS1-specific IgG antibody responses to 61 DENV1 NS1 overlapping peptides in patients with DF (n = 10) and DHF (n = 10) were measured by ELISA on day 7 of illness. Differences in mean values between antibody responses to overlapping peptides in patients with DF and DHF were compared using the Mann–Whitney U test (two tailed). NS1 antibody levels in patients with DHF are indicated in red, and in those with DF in blue. Error bars indicate mean and standard deviation (SD). *P < 0.05, **P < 0.01, ***P < 0.001

(JEV) virus and West Nile virus (WNV) with all four serotypes of the DENVs using clustal omega[29], as these are the only two reported flaviviruses co-circulating in Sri Lanka (Supplementary Fig. 4)[36]. We found that the majority of NS1 antibody epitopes that we have described in this study, were directed against the regions that were highly conserved among DENV1, 2, 3 and 4,

JEV and WNV (Supplementary Fig. 4a). The regions were similarly conserved when comparing the WNV, JEV and NS1 sequences of DENV1 and DENV2 strains that recently circulated in Sri Lanka (Supplementary Fig. 4b). Therefore, the majority of NS1 antibodies defined in this study are likely to be highly cross-reactive.

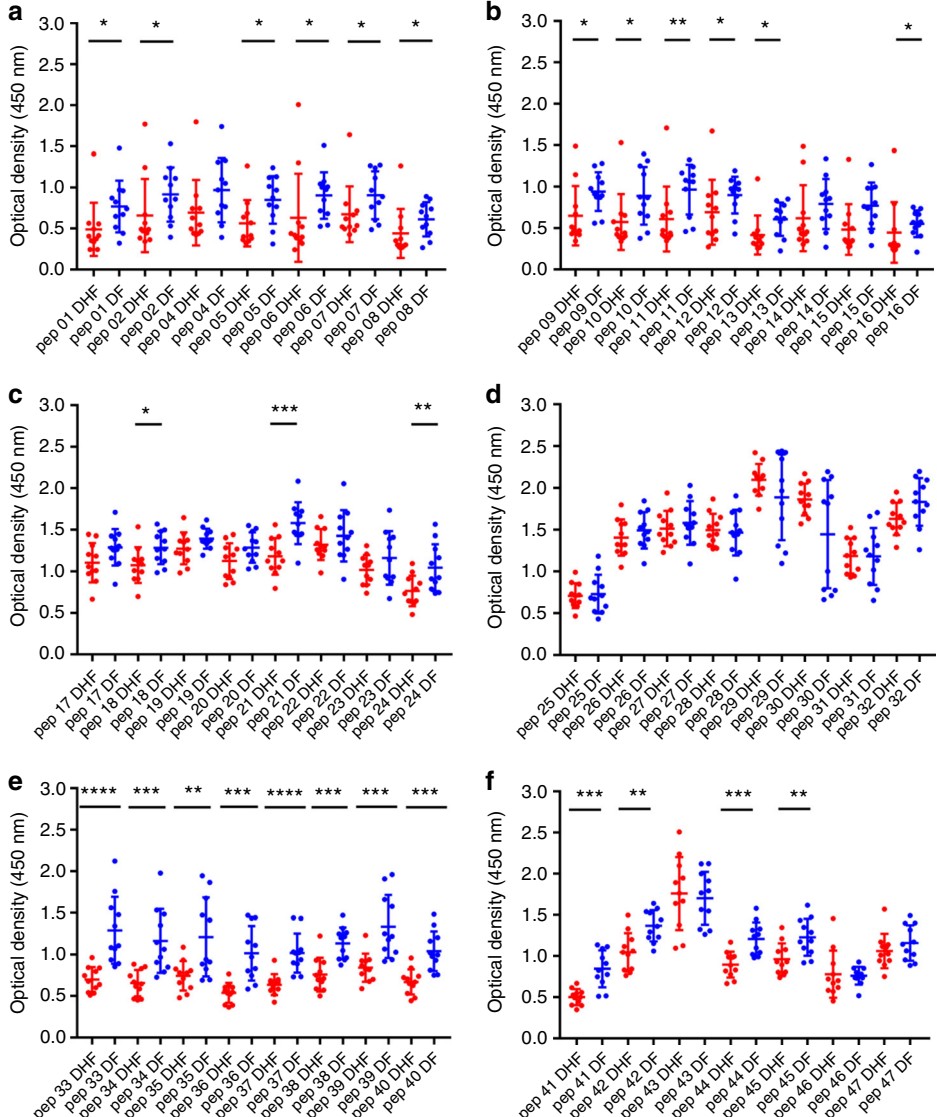

**Fig. 6** DENV2 NS1 antibody responses to NS1 peptides in the critical phase. **a–f** DENV2 NS1-specific IgG antibody responses to 47 DENV2 NS1 overlapping peptides were measured by ELISA in patients with DF ($n = 10$) and DHF ($n = 10$) on day 3 of illness. Differences in mean values between antibody responses to overlapping peptides in patients with DF and DHF were compared using the Mann–Whitney U test (two tailed). Error bars indicate the means and standard deviation (SD). *$P < 0.05$, **$P < 0.01$, ***$P < 0.001$, ****$P < 0.0001$

## Discussion

In this study, we sought to improve our understanding of the specificity of NS1 antibody responses in mild and severe dengue infection disease states. We found that NS1 antibody levels were significantly higher in healthy individuals who had past SD when compared with those who had past non-severe dengue (NSD). In patients with acute infection, the NS1 antibodies rose more quickly and to a greater extent in patients with DHF during acute secondary dengue infection in the critical phase of illness compared with those with DF. Interestingly, disease severity was reflected in the epitope bias of the antibody response; patients with DF and DHF displayed antibody responses to distinct regions of DENV–NS1 protein, as defined by the constitutive peptide fractions. These differences among patients with DF and DHF were observed during the febrile phase of the illness (before the onset of vascular leak in any patients) as well as during the critical phase. Although there were some differences in NS1 antibody epitopes recognized by

patients with an acute secondary DENV1 infection compared with those with DENV2 infection, in both cohorts, the antibody responses were directed at the conserved regions of NS1. However, we only assessed NS1 antibody responses in acute DENV1 and DENV2 infection and therefore, it is difficult to predict if similar observations would be made in patients with acute DENV3 or DENV4 infection. When we further characterized NS1 antibody responses in a cohort of healthy individuals who either had past DHF (SD) or NSD, we found similarly that participants with past SD and NSD mounted antibody responses that recognized different regions of NS1, when measured ex vivo to NS1 peptides. Interestingly, those with milder forms of acute dengue and past NSD generated antibody responses to shared regions of NS1, distinct to the epitopes recognized by antibody responses of those with current or past severe dengue. This suggests that antibody recognition of certain epitope regions of dengue NS1 protein could confer a greater protective response to dengue infection.

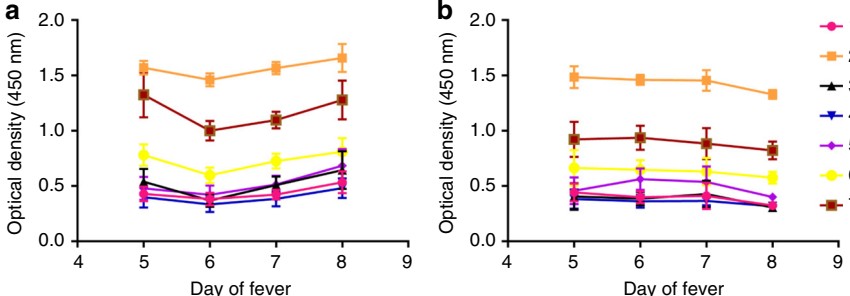

**Fig. 7** NS1 antibody responses to selected regions of NS1. Antibody levels were measured in ten patients with DF (**a**) and ten patients with DHF (**b**), with acute secondary DENV1 infection throughout the course of illness to seven pools of selected NS1 overlapping peptides. Pool 1 represents aa 1–33, pool 2 aa 121–142, pool 3 aa 161–195, pool 4 aa 222–249, pool 5 aa 251–284, pool 6 aa 279–307 and pool 7 aa 341–353. The bars indicate the means and standard error of mean (SEM)

A previous study characterized NS1 antibody epitopes in ten children with DF, seven children with DHF and three with DSS, which included children with an acute primary or secondary dengue infection with DENV1, DENV2 and DENV3 serotypes, and used a similar approach as us. It was shown that the most immunodominant regions of NS1 were those represented by aa 101–121, aa 111–131 and aa 301–321[37]. In our patient cohort, we found somewhat overlapping regions with the peptides that were identified in the previous study in paediatric patients with acute secondary DENV1 (aa121–142 and aa 291–307) and in acute secondary DENV2 (aa 297–313). However, most of the NS1 antibodies identified by us were directed at different regions. The reasons for these differences could be due to our dengue cohorts being patients with secondary dengue whereas, Hertz et al. had mapped epitopes in patients with both primary and secondary infection. In addition, we examined the antibody epitopes in 20 adult patients with DENV1 and 22 patients with DENV2, whereas Hertz et al. had mapped antibody epitopes in 27 children with DENV2 ($n = 7$) and DENV3 ($n = 14$), and therefore the variations in the sample size, ethnicity and age could have contributed to these differences.

Many of the studies that show NS1 antibodies are associated with protection have been carried out in dengue mouse models[15,38]. Proteomics, sequence identity analyses and in vitro studies have shown that the C-terminal region of NS1 contains many cross-reactive epitopes, with sequence homology to self-antigens such as endothelial cell proteins and mediators involved in the coagulation pathways and platelet response[9,39–41]. As a result, the monoclonal antibody, which substitutes the C-terminus of dengue NS1 with that of Japanese Encephalitis NS1 was found to be protective in dengue mouse models[42]. Our data show that DF patients with acute secondary DENV1 and DENV2 infection, had significantly higher antibody titres to the C-terminus of NS1 compared with those with severe forms of dengue (DHF). In fact, when we studied the antibody response kinetics to the C-terminus in patients with DHF and DF, we observed that the levels of antibody directed at the C-terminus of NS1, rose significantly preceding the recovery phase of DF patient illness, whereas DHF responses had declined by this time point. In addition, healthy participants who experienced past NSD also had significantly higher antibody titres directed towards the C-terminus of NS1. Therefore, although the C-terminus appears to share cross-reactive regions with self-proteins, milder forms of illness associate with higher antibody titres targeted to this region. However, it is noteworthy that in our experiments, we used NS1 overlapping peptides which might not reflect the actual binding of NS1 antibodies to the fully folded and polymeric structure. This indeed may account for our observation that antibody responses

to the whole recombinant NS1 protein are greater in DHF, whereas we detected higher antibody responses to NS1 peptides in serum of patients with DF. Therefore, in order to further characterize the possible protective and pathogenic antibodies to NS1, it would be important to map antibody epitopes to the dimeric and hexameric forms of NS1.

Although we observed similarities in the NS1 antibody responses from patients with acute secondary DENV1 and DENV2, we did observe some interesting differences. For instance, the difference in NS1 antibody titres during the critical phase between patients with DF and DHF was more marked during DENV1 infection. In addition, while those with DF, due to either serotype, had higher antibody titres to the C-terminus region of NS1, DHF patients with DENV2 infection had significantly higher antibody responses directed towards the N-terminus. Although all DENV serotypes are known to cause severe disease, the severity of infection and clinical features have been shown to vary depending on the viral serotype and genotype[27,43,44]. During this study period, it was found that patients with DENV2 had a higher risk of developing DHF, and more rapid progression to plasma leakage[27]. Therefore, the targets of NS1 antibodies generated to each serotype, may play a role in disease pathogenesis.

It has been shown that NS1 antibodies bind to and result in destruction of platelets in dengue mouse models[45,46]. Our data showed a trend towards inverse correlation between NS1 antibody levels and platelet counts in patients with DHF. Although the possibility that NS1 antibodies lead to thrombocytopenia in humans has not been investigated, one possible explanation for our observations is that NS1 antibodies could be binding to platelets in patients with DHF, causing their elimination. Since the NS1 antibody repertoire appears to be different in those with DF compared with DHF, it is possible that patients with DHF may have NS1 antibodies that preferentially bind platelets. Therefore, careful experiments should be carried out to determine whether such pathways may contribute to disease pathogenesis.

Dengue mouse models have suggested that NS1 antibodies were associated with protection, yet human in vitro experiments have cast some doubt on this hypothesis. Studies of human sera from those with mild and severe forms of dengue have shown that the binding of NS1 antibodies to endothelial monolayers was significantly enhanced when using sera from patients with severe dengue during the critical phase of illness, a possible route to inducing life-threatening plasma leak[9]. Furthermore, NS1–NS1 antibody complexes have been shown to activate complement[11] and these complexes are known to cross-react with platelets, which could be linked to the haemorrhage seen in severe disease. Therefore, it has been speculated that NS1 antibody complexes

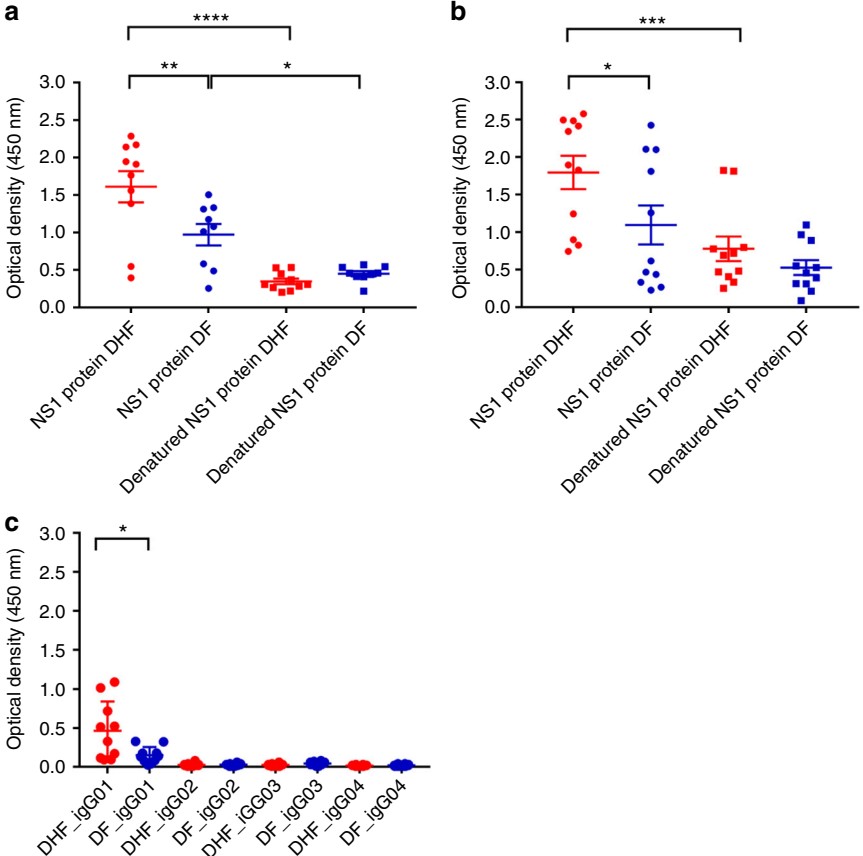

**Fig. 8** DENV NS1 antibody responses to recombinant and denatured NS1. **a** DENV1 NS1-specific IgG antibody responses to recombinant DENV1 NS1 and denatured DENV1 NS1 were measured by ELISA in patients with DF ($n = 10$) and DHF ($n = 10$) on day 7 of illness. **b** DENV2 NS1-specific IgG antibody responses to recombinant DENV2 NS1 and denatured DENV2 NS1 were measured by ELISA in patients with DF ($n = 10$) and DHF ($n = 10$) on day 6 of illness. **c** DENV2 NS1-specific IgG subclasses to DENV-2 NS1 protein were measured in patients with DF ($n = 10$) and DHF ($n = 10$) on day 6 of illness. Differences in mean values between antibody responses to NS1 protein in patients with DF and DHF were compared using the Mann–Whitney U test (two tailed). Error bars indicate mean and standard deviation (SD). *$P < 0.05$, **$P < 0.01$, ***$P < 0.001$, ****$P < 0.0001$

are involved in disease pathogenesis[41,47]. Our data show that NS1 antibodies become elevated in patients with DHF earlier than DF. Since the preferential NS1 antibody binding epitopes are different in patients with DHF and DF (and in those with past SD and NSD), it is possible that patients with DHF develop antibodies to certain regions of the NS1, which are associated with enhanced complement activation and endothelial dysfunction, whereas patients with DF produce NS1 antibodies directed against regions of NS1 that neutralize the pathogenic action of NS1. Therefore, it would be crucial to study monoclonal antibodies generated in patients with varying severity of illness during acute infection and assess the ability of these antibodies to activate complement or to neutralize the pathogenic effects of NS1.

The NS1 protein of the flaviviruses is highly conserved[35] and therefore, antibodies directed at the NS1 of a particular flavivirus are likely to cross-react with other flavivirus NS1. Most of the NS1 antibody epitopes defined in this study and which were found to be significantly increased in patients with DF are directed to these highly conserved regions of NS1 and had a cross-reactivity of >65% with JEV and WNV. Therefore, it is possible that these NS1 antibodies were generated due to infection with another flavivirus, that subsequently expanded in infection with DENV. It would be important to test the NS1 antibody responses in those who had received the JEV vaccine but are negative for DENV to define their NS1 antibody repertoire to determine such cross-reaction.

In summary, our data show that NS1 antibody titres rise in patients with acute secondary dengue with the onset of the critical phase of disease. Notably anti-NS1 antibody production rises more quickly and to a greater extent in patients with the most severe dengue infection. In addition, antibodies from patients with more severe forms of acute illness and those who have had past DHF recognize different epitope regions of NS1 protein compared with those from patients who have or have previously experienced milder forms of acute dengue. These data suggest that patients with more severe forms of dengue could have a qualitatively and quantitatively different anti-NS1 antibody repertoire, which contributes to disease severity. These findings have implications for approaches to treatment and vaccination.

## Methods

**Patients.** Adult patients with varying severity of acute dengue infection were recruited from the National Institute of Infectious Disease, Sri Lanka following informed written consent. Daily consecutive blood samples were obtained from 76 patients from the day of admission to the hospital until they were discharged from the hospital, capturing the course of illness. The day on which the patient first developed fever was considered day 1 of illness. Only those whose duration from onset of illness was ≤ 4 days were recruited. All clinical features, including the presence of fever, abdominal pain, vomiting, bleeding manifestations, hepatomegaly, blood pressure, pulse pressure and evidence of fluid leakage, were recorded several times each day. Fluid leakage was assessed by ultrasound scans to detect fluid in pleural and peritoneal cavities. Clinical disease severity was classified according to the 2011 World Health Organization (WHO) dengue diagnostic criteria[23]. Patients with a rise in haematocrit > 20% of baseline or clinical or

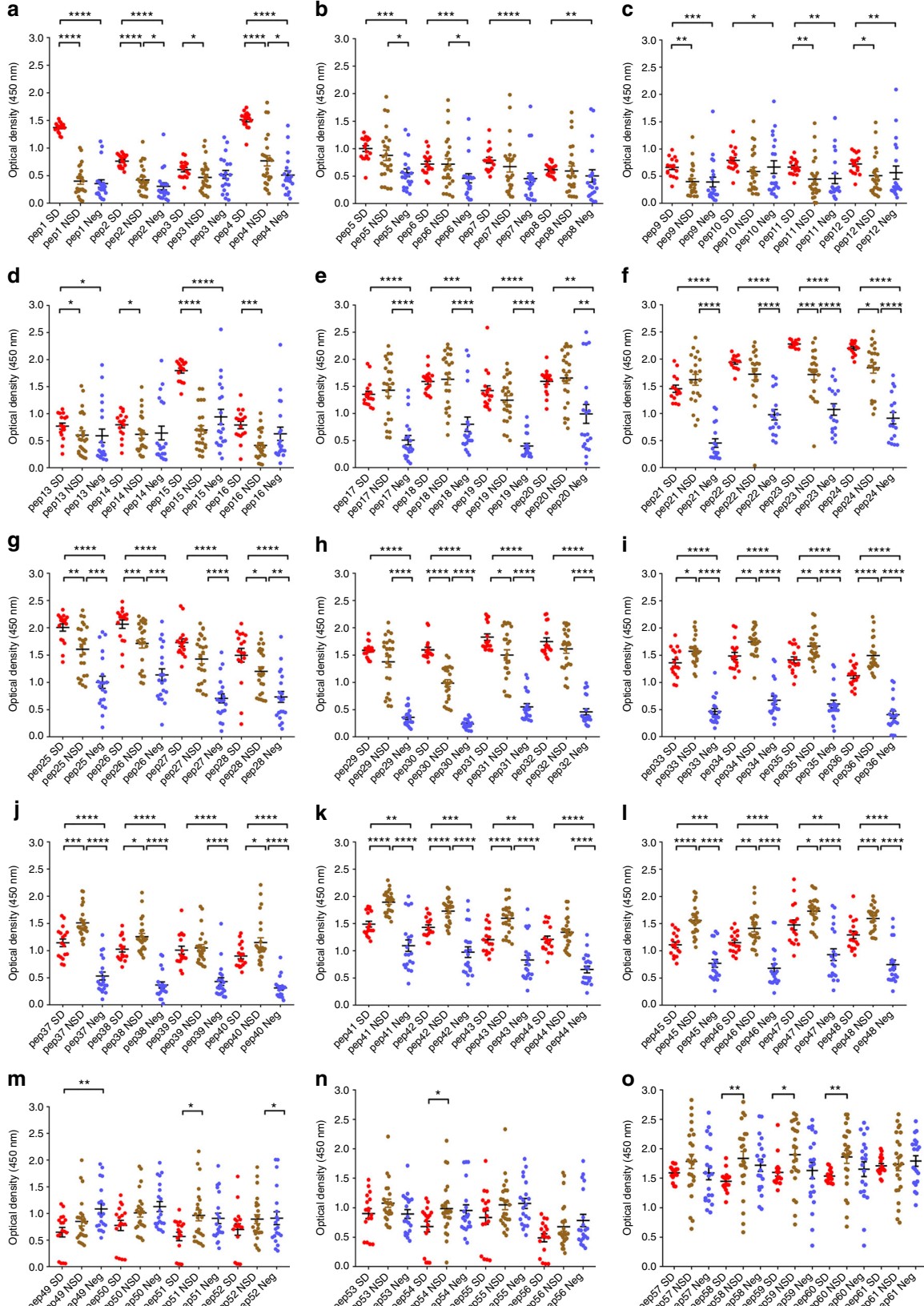

**Fig. 9** NS1-antibody responses in individuals with past infection. **a–o** DENV1 NS1-specific IgG antibody responses to DENV-1 NS1 overlapping peptides 1–61 in seronegative individuals (Neg, $n = 20$), those with past non-severe dengue (NSD, $n = 24$) and past severe dengue (SD, $n = 17$). Error bars represent mean and standard deviation. Differences between the groups were compared using the Mann–Whitney U test (two tailed). *$P < 0.05$, **$P < 0.01$, ***$P < 0.001$, ****$P < 0.0001$

ultrasound scan evidence of plasma leakage were classified as having DHF. Shock was defined as having cold clammy skin, along with a narrowing of pulse pressure of 20 mmHg. As such, 36 patients were classified as DHF and 40 patients were classified as DF.

**Healthy individuals with varying severity of past dengue**. In our previous studies, we had recruited 1689 healthy individuals attending the Family Practice Centre, which is a primary health care facility of the University of Sri Jaye-wardenepura, Sri Lanka, providing community healthcare to over 2000 families living in the suburban areas of the Colombo district[48]. In this cohort, we have serum samples from the time of recruitment to the study, information regarding dengue serostatus and whether the individuals had DHF or mild or asymptomatic dengue. Seventeen serum samples were used from those who reported an episode of DHF (past severe dengue (SD)), and 24 serum samples were used from those who were seropositive for dengue but had never been hospitalized due to a febrile illness. The individuals who were seropositive but had never been hospitalized due to any febrile illness were considered to have a mild/inapparent or non-severe dengue infection (past non-severe dengue infection (NSD)). We also used 20 serum samples from individuals who were seronegative at the time of recruitment. Only one of them had received the Japanese encephalitis vaccine, and all individuals (including the person who received the JEV vaccine) were seronegative for JEV.

**Ethics statement**. Ethical approval was obtained by the Ethics Review Committee of the Faculty of Medical Sciences, University of Sri Jayawardenapura. All patients were recruited following informed written consent. We have complied with all relevant ethical regulations in carrying out this study.

**Serotyping of DENV and assessment of viral titre**. Acute dengue infection was confirmed by quantitative real-time PCR, and DENV viruses were serotyped and titres quantified by quantitative real-time PCR[28]. Viral RNA in serum was extracted using QIAamp Viral RNA Mini Kit (Qiagen, USA, Cat: 52906) and transcribed to cDNA using High Capacity cDNA reverse transcription kit (Applied Biosystems, USA, cat: 4368814) as per the manufacturer's protocol in 20 -μl reaction mixtures containing $1 \times$ RT buffer, $1 \times$ RT random primers, $10 \times$ dNTP Mix (100 mM), 50 U of MultiScribe™ Reverse Transcriptase, 20 U of RNase Inhibitor (Applied Biosystems, USA, Cat: N8080119), 10 μl of RNA and PCR-grade water (Applied Biosystems, USA, Cat: AM9935).

Multiplex quantitative real-time PCR was performed using the CDC real-time PCR assay for detection of the dengue virus[49], and modified to quantify DENV. Oligonucleotide primers and a dual-labelled probe for DEN 1, 2, 3, 4 serotypes were used (Life Technologies, India) based on published sequences[49]. In order to quantify viruses, a tenfold dilution series ($10^6$–$0^0$ PFU/ml) was prepared using the cDNA of reference strains for standard curves[28]. Real-time PCR was performed using TaqMan® Multiplex Master Mix (Applied Biosystems, USA, Cat: 4461881). The reactions consisted of 20- μl volumes and contained the following reagents: $1 \times$ TaqMan multiplex master mix (containing Mustag Purple dye), 900 nM of each primer, 250 nM of each probe, 2 μl of cDNA and PCR-grade water (Applied Biosystems, USA, Cat: AM9935). Following initial denaturation for 20 s at 95 °C, the reaction was carried out for 40 cycles of 3 s at 95 °C and 30 s at 60 °C. The threshold cycle value (Ct) for each reaction was determined by manually setting the threshold limit. All assays were done in triplicate. After the primers and probes were validated, a multiplex method was optimized to quantify the four serotypes in a single reaction and was used to determine viral loads of unknown samples that were determined using the optimized multiplex real-time protocol.

**Analysis of dengue NS1 antigen and DENV IgM and IgG levels**. The presence of NS1 antigen was assessed using the NS1 early dengue enzyme-linked immuno-sorbent assay (ELISA) (Panbio, Brisbane, QLD, Australia). NS1 antigen levels were semi-quantitatively assessed in serial blood samples of patients and expressed as Panbio units. Dengue antibody assays were performed using a commercial capture-IgM and IgG ELISA (Panbio, Brisbane, Australia)[50,51]. Based on the WHO criteria, patients with an IgM:IgG ratio of > 1.2 were considered to have a primary dengue infection, while patients with IgM:IgG ratios < 1.2 were categorized under sec-ondary dengue infection[52]. The DENV-specific IgM and IgG ELISA was also used to semi-quantitatively determine the DENV-specific IgM and IgG titres, which were expressed as Panbio units.

**Development of an ELISA to detect NS1-specific antibody**. Currently, there are no assays to measure NS1 antibody levels, and as such, we developed an ELISA to measure NS1 antibody levels in patient samples. Commercially available recom-binant DENV1 and DENV2 NS1 full-length recombinant proteins (Native antigen, USA) expressed from a mammalian cell line were used to coat the ELISA plates and capture NS1-specific antibodies in serum samples. The optimal concentration of coating antigen, serum dilution and concentration of secondary antibody were determined by checkerboard titrations.

Our indirect ELISA to detect dengue NS1-specific antibodies was performed in 96-well microtitre plates (Pierce™, Cat: 15031). The plates were coated with either 100 μl/well DENV1 NS1 protein (Native antigen, USA) or DENV2 NS1 protein (Native antigen, USA) diluted in carbonate–bicarbonate coating buffer (pH 9.6) at

a final concentration of 1 μg/ml and incubated overnight at 4 °C. After incubation, the wells were washed three times with 300 μl/well washing buffer (phosphate-buffered saline (PBS) containing 0.05% Tween 20). The wells were blocked with 300 μl/well with blocking buffer (PBS containing 0.05% Tween 20 and 1% bovine serum albumin (BSA)) for 1 h at room temperature. The wells were washed again three times with 300 μl/well washing buffer. Serum samples were diluted 1:5000 in PBS containing 0.05% Tween 20 and 1% BSA. Diluted serum samples were added in triplicates to the appropriate plate at 100 μl/well and incubated for 60 min at room temperature. After incubation, wells were washed three times with 300 μl/well washing buffer. Goat anti-human IgG, biotinylated antibody (Mabtech, Sweden, Cat:3820-4-250) was diluted to 1:1000 in PBS containing 0.05% Tween 20 and 1% BSA and added 100 μl/well. The plates were incubated again for 30 min at room temperature and washed three times with washing buffer. Streptavidin alkaline phosphatase (Abcam, UK, Cat: ab64268) was diluted 1:20 with PBS containing 0.05% Tween 20 and 1% BSA and added 100 μl/well. The plate was incubated for 30 min at room temperature and washed five times. Para-nitrophenylphosphate (PNPP) (Thermo Fisher Scientific, USA, Cat: 37621) substrate was added at 100 μl/well and incubated for 20 min in the dark at room temperature. The reaction was stopped by adding 50 μl/well 2 M NaOH, and the ELISA read on MPSCREEN MR-96A ELISA reader at 405-nm wavelength. Sample diluent (PBS containing 0.05% Tween 20 and 1% BSA) was used as the blank for each assay. Using this optimized assay, we determined NS1 antibody levels in DENV1 DF ($n = 10$), DENV1 DHF ($n = 10$) and DENV-seronegative individuals ($n = 20$). The assay showed a significant difference in NS1 antibody levels between both DHF and seronegative individuals ($p < 0.0001$) and DF and seronegative individuals ($p = 0.0302$). Furthermore, we determined NS1 antibody levels in DENV2 DF ($n = 10$), DENV2 DHF ($n = 17$) and DENV-seronegative individuals ($n = 18$). Again, the assay showed significant differences between both DHF and seronegative individuals ($p < 0.0001$) and DF and seronegative individuals ($p = 0.0001$). Healthy individuals with past severe dengue ($n = 34$) and past non-severe dengue ($n = 36$) too resulted in a significantly different antibody level ($p < 0.0001$) compared with seronegative individuals ($n = 20$).

**NS1 antibody responses to DENV1 and DENV2 overlapping peptides**. Dengue NS1 peptide arrays were obtained through the NIH Biodefense and Emerging Infections Research Resources Repository, NIAID (NIH: Peptide Array, Dengue Virus Type 1, Singapore/S275/1990, NS1 Protein, NR-2751) (Accession code: P33478 and NIH: Peptide Array, Dengue Virus Type 2, New Guinea C (NGC), NS1 Protein, NR-508) (Accession code: AAA42941). Ninety-six-well microtitre plates (Pierce™ Cat:15031) were coated with 100 μl/well peptide preparations diluted in bicarbonate/carbonate coating buffer (pH 9.6) at a final concentration of 1 μg/100 μl. The peptides were coated individually and incubated overnight at 4 °C. After incubation, unbound peptide was washed away twice with phosphate-buffered saline (PBS, pH 7.4). The wells were blocked with 250 μl/well PBS con-taining 0.05% Tween 20 and 1% bovine serum albumin (BSA) and incubated for 1.5 h at room temperature. The plates were then washed three times with 300 μl/well washing buffer (PBS containing 0.05% Tween 20). Serum samples were diluted 1:250 in ELISA diluent (Mabtech, Sweden, Cat: 3652-D2). Diluted serum samples were added 100 μl/well in duplicates and incubated for 30 min at room tem-perature. After incubation, wells were washed three times with 300 μl/well washing buffer. Goat anti-human IgG, biotinylated antibody (Mabtech, Sweden Cat: 3820-4-250) was diluted 1:1000 in PBS containing 1% BSA and added 100 μl/well. Plates were then incubated for 30 min at room temperature and washed three times as described previously. Streptavidin–HRP (Mabtech, Sweden, Cat: 3310-9) was diluted 1:1000 in PBS containing 1% BSA and added 100 μl/well. Incubation was carried out for 30 min at room temperature, and the washing procedure was repeated five times. TMB ELISA substrate solution (Mabtech, Sweden, Cat: 3652-F10) was added 100 μl/well and incubated in the dark for 10 min at room tem-perature. The reaction was stopped by adding 100 μl/well 2 M $H_2SO_4$ stop solution and absorbance values were read at 450 nm.

**Analysis of antibody binding to NS1 epitopes**. DENV1 NS1 and DENV2 NS1 recombinant proteins (Native antigen, USA) were added into separate wells in carbonate–bicarbonate coating buffer (pH 9.6) at a final concentration of 1 μg/ml and were incubated overnight at 4 °C. NS1 proteins, which were denatured by heating them for 95 °C for 15 min[30], were used to compare the responses to the recombinant NS1 in this assay. The wells were blocked with 300 μl of blocking buffer (PBS containing 1% bovine serum albumin (BSA)) for 1 h at room tem-perature. DENV1 and DENV2 serum samples were diluted 1:5000 with ELISA diluent (Mabtech, Cat: 3652-D2). Diluted serum samples were added at 100 μl/well into respective wells and incubated for 60 min at room temperature. The bound NS1-specific antibodies were detected using 100 μl/well goat anti-human IgG, biotinylated antibody (Mabtech, Sweden, Cat: 3820-4-250), diluted 1:1000 in PBS containing 1% BSA and incubated for 30 min at room temperature. Streptavidin–alkaline phosphatase (Abcam, UK, Cat: ab64268) diluted 1:20 in PBS containing 1% BSA was added 100 μl/well and incubated for 30 min at room temperature. PNPP (Thermo Fisher Scientific, USA, Cat: 37621) substrate solution was added 100 μl/well and incubated in the dark for 20 min at room temperature. The reaction was stopped by adding 100 μl/well 2 M NaOH stop solution, and absorbance values were read at 405 nm.

**Characterization of DENV2 NS1 IgG subtype responses**. DENV2 NS1 recombinant protein was bound to microplate wells with carbonate–bicarbonate coating buffer (pH 9.6) at a final concentration of 1 μg/ml and was incubated overnight at 4 °C. The wells were blocked with 300 μl of blocking buffer (PBS containing 1% bovine serum albumin (BSA)) for 1 h at room temperature. DENV2 serum samples diluted 1:5000 in ELISA diluent (Mabtech, Sweden, Cat: 3652-D2) were added at 100 μl/well and incubated for 60 min at room temperature. The bound NS1-specific IgG antibody subtypes were detected using 100 μl/well goat anti-human IgG1 (Mabtech, Sweden, Cat: 3851-6-250), IgG2 (Mabtech, Sweden, Cat: 3852-6-250), IgG3 (Mabtech, Sweden, Cat: 3853-6-250) and IgG4 (Mabtech, Sweden, Cat: 3854-6-250) biotinylated antibody subtypes separately, diluted 1:1000 in PBS containing 1% BSA. Plates were then incubated for 30 min at room temperature and streptavidin–alkaline phosphatase (Abcam, UK, Cat: ab64268) diluted to 1:20 in PBS containing 1% BSA was added at 100 μl/well for 30 min at room temperature. PNPP (Thermo Fisher Scientific, USA, Cat: 37621) substrate solution was added at 100 μl/well and incubated in the dark for 20 min at room temperature. The reaction was stopped by adding 100 μl/well 2 M NaOH stop solution and absorbance values were read at 405 nm.

**Statistical analysis**. Statistical analysis was performed using GraphPad Prism version 6. As the data were not normally distributed (as determined by the frequency distribution analysis of GraphPad PRISM), non-parametric tests were used in the statistical analysis and two-sided tests were carried out in all instances. Differences in the serial values of NS1 antibodies, NS1 antigen and viral loads in patients with DHF and DF were calculated/compared using the Holm–Sidak method. Corrections for multiple comparisons were completed using the Holm–Sidak method, and the statistical significant value was set at 0.05 (alpha). Differences in mean values between antibody responses to overlapping peptides in patients with DF and DHF and in individuals with SD and NSD were compared using the Mann–Whitney U test (two tailed).

## Data availability

All data generated or analyzed during this study are available included in the main manuscript and the supplementary files. All data are available without restrictions.

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

## Acknowledgements

Funding was provided by the Centre for Dengue Research, University of Sri Jaye-wardenapura, National Research Council, Sri Lanka (15-043), National Science Foundation, Sri Lanka (RPHS/2016/D-06) and by the Medical Research Council (UK). Graham Ogg receives support from the National Institute for Health Research (NIHR) Oxford Biomedical Research Centre (BRC).

## Author contributions

G.N.M. and G.S.O. designed the project and experiments, D.J. and L.G. performed the NS1 antibody experiments, S.F., L.G. and C.J. did the viral loads and other antibody experiments, C.J., P.G.S.B.J., D.H., M.A.P.A.P., S.F. and A.W. recruited the patients and collected all clinical data, D.J., L.G. and G.N.M. analyzed and interpreted the data and D.J., G.N.M., C.S.H. and G.S.O. wrote the paper.

## Additional information

**Competing interests:** G.N.M. is a consultant to Sun Pharma Industries Ltd. The remaining authors declare no competing interests.

