## [Peer Review File · Nature Communications]

Reviewers' comments:

Reviewer #1 (Remarks to the Author):

Jayathilaka et al. reported on the role of NS1 antibodies in the pathogenesis of acute dengue virus infection. The causing factors for dengue hemorrhagic fever (DHF) are an enigma as of today. The authors attempted to explain that NS1 antibodies contributed to be one of the causing factors. Their evidence was tabulated by recruiting acute dengue patients and checked for the presence of NS1 antibodies and its association with dengue severity and the antibody interacting with specific element of the NS1 could play a significant role in pathogenesis. Although it is important to understand and dissect the mechanism leading to DHF, the cumulated NS1 results in current manuscript could not persuade this reviewer yet.

The major concerns are:

1. The stage of dengue virus infection in a subject is a time dependent, and as of Sabin's study in human being in 1952, which he clearly demonstrated that an incubation time in infected subjects were 5-7 days, followed by febrile and viremia stage, so did the appearance of NS1 protein in parallel, and then defebriate and clear off virus as well as protein stage, a period that a rash or hemorrhagic occurred in affected subjects. The kinetic of the clinical manifestations in dengue patient is well-defined. As such, if a subject who is secondary infection, we can envision that anamnestic antibody response should occur during the incubation period due to the recall event. In this stage, the antibody, either specific to dengue virus or NS1 protein, should be detected. Hence, DHF should be observed during early phase of the febrile rather than during the defebriate stage. As such, the author should perform the basal line levels of the NS1 specific antibody, especially to the N-terminal peptide, in general healthy population prior to draw the conclusion.
2. There are many risk factors involved in development of DHF. One of such factors was addressed by the author, secondary dengue virus infection based upon the IgM: IgG ratio. This is the major flaw of the definition. As of today, there has been reported that some of infected subjects, the IgM levels can be lasted for quite some times, that some of these subjects may not have response to the infection, and that some protected individuals are low in IgG levels. As such, the authors may need to perform the T cells, especially the memory T cells, to sort out the issue.

Minor points:

1. The title of the manuscript is too broad and does not fit the contents in the manuscript.
2. Page 6, a reference should be added after the statement. "In DENV1-infected DHF patients the serum concentration of NS1 antibodies started to rise 5.5 days after the onset of illness." This sentence is misleading since the onset of illness means fever, which has 5.6 days of incubation time already. It is important to know that the recall antibody response is earlier in secondary dengue virus infected subjects. As such, the sentence required to be rephrased to reflect the actual time line since the time of virus inoculation.
3. The levels of dengue NS1 protein in circulation of the acute patient have been reported to be correlated with the disease severity. In current manuscript, the authors did not report the concentration of NS1 protein in these recruited patients. Importantly, the NS1 proteins in dengue patients has been shown to be hexamer formation, did authors check the NS1 antibodies bind to the NS1 harvested from acute patients? The critical questions will be "will the higher the concentration of NS1 protein result in higher levels of NS1 antibodies, and subsequent lead to severe dengue?" In addition, the viral load in these recruited patients have to be measured and presented.
4. Since other flaviviruses may circulate in the country, it is important to clarify whether these specific NS1 antibody is not due to cross-reactivity of other viruses within the family. In addition, how close the NS1 protein in dengue virus to that of Zika virus in sequence alignment? Furthermore, did the authors check whether these recruited subjects had any autoimmune status?
5. In methods, the authors addressed that "Only those whose duration from onset of illness was ≤4 days were recruited." But in the presented figure, all were shown more than 5 days of illness.

Please clarified.

Reviewer #2 (Remarks to the Author):

In their manuscript, Jayathilaka et al. addressed the role of anti-NS1 antibodies in the pathogenesis of secondary dengue infections. Authors studied the anti-NS1 antibody kinetics in DF and DHF patients with secondary DENV-1 and DENV-2 infections and explored the anti-NS1 repertoires. Their results suggest that the epitope targeted by the antibody response could play an important role in the dengue pathogenesis and could lead to better vaccine and treatment design. These results are partially novel and of high interest in the field of dengue pathogenesis. The paper is clear, well-written and globally scientifically sound. Nevertheless, I do have some concerns that I would like the authors to address.

Major comments:

- People with past-asymptomatic dengue infection is one of the group studied here and I have some concerns with that. It is already very complex in a prospective study, with close monitoring, to identify dengue-infected people who remain asymptomatic, and I don't really understand how it is possible to ensure that people had an asymptomatic form of dengue using the health care facility as described in the M&M section? If the dengue episode was pauci-symptomatic or mild, most people may simply not recall the event after few weeks or months at the time they are tested at the health facility. It is therefore most likely that the group of individuals named "asymptomatic" or "inapparent" is a mix of people who experienced truly asymptomatic, pauci-symptomatic and mild disease. From a pathophysiological perspective, being asymptomatic and developing a mild form of the disease (most frequent form during primary infection in children) is not really the same. I would like the authors to clarify that point and since it may not be possible to include a real "asymptomatic" group to discuss how this may impact globally their conclusions.
- Results should be compared to those generated by the study of Hertz T et al., published in J Immunol (available online on April 2017) as these authors previously looked at the role of anti-NS1 antibodies with sometimes a similar approach. Interpreting the results of this study in the light of what Hertz already described would provide a better understanding of the complex mechanisms involved in dengue pathogenesis.
- Authors should provide a little bit more alternative explanation regarding the differences in NS1 antibody kinetic between DENV-1 and DENV-2, e.g. the serotypes involved in previous infections/sequence of previous infections; the variations between different strains of the same serotype on NS1 production (Watanabe, J Virol 2012), etc.

Minor comments:

- Title should be "Role of NS1 antibodies in the pathogenesis of acute secondary dengue infections" (no primary cases were included in the study)
- Please be consistent with the use of DENV rather than DEN when referring to the dengue virus
- Line 47: this sentence should start with "In addition" rather than "Indeed" as the increased likelihood of disease enhancement is not related to the 2 issues mentioned in the previous sentence (poor efficacy in naive patients and efficacy depending on serotype)
- Line 49: NS1 is also a marker of dengue infection (can be found in asymptotically-infected individuals), not only of disease
- Line 191: "p=" is missing for the peptide pools 2
- Line 312: What about impact on platelet counts observed in DF patients with DENV-2 infection?

- Discuss briefly the result limitations since only DENV-1 and DENV-2 infections were studied
- For the in-house assays developed for the purposes of this study, more details on how the threshold were set (positive and negative controls), the wavelengths used by the reader, if samples were tested in duplicates, triplicates, etc. would help the reader to develop the assays but also to better trust the results.

Reviewer #1 (Remarks to the Author):

Jayathilaka et al. reported on the role of NS1 antibodies in the pathogenesis of acute dengue virus infection. The causing factors for dengue hemorrhagic fever (DHF) are enigma as of today. The authors attempted to explain that NS1 antibodies contributed to be one of the causing factors. Their evidence was tabulated by recruiting acute dengue patients and checked for the presence of NS1 antibodies and its association with dengue severity and the antibody interacting with specific element of the NS1 could play a significant role in pathogenesis. Although it is important to understand and dissect the mechanism leading to DHF, the cumulated NS1 results in current manuscript could not persuade this reviewer yet.

The major concerns are:

1. The stage of dengue virus infection in a subject is a time dependent, and as of Sabin's study in human being in 1952, which he clearly demonstrated that an incubation time in infected subjects were 5-7 days, followed by febrile and viremia stage, so did the appearance of NS1 protein in parallel, and then defebriile and clear off virus as well as protein stage, a period that a rash or hemorrhagic occurred in affected subjects. The kinetic of the clinical manifestations in dengue patient is well-defined. As such, if a subject who is secondary infection, we can envision that anamnestic antibody response should occur during the incubation period due to the recall event. In this stage, the antibody, either specific to dengue virus or NS1 protein, should be detected. Hence, DHF should be observed during early phase of the febrile rather than during the defebriile stage. As such, the author should perform the basal line levels of the NS1 specific antibody, especially to the N-terminal peptide, in general healthy population prior to draw the conclusion.

Response: We thank the reviewer for this very important comment, which we agree would be to help understand how NS1 antibodies contribute to either DF or DHF. We have now assessed the NS1 antibody epitopes in a general healthy population and also we carried out additional experiments in the patient cohort described in the manuscript at baseline (at the time of recruitment), before they developed DHF. As suggested by the reviewer, we investigated the baseline NS1 protein specific responses in healthy individuals who were dengue seropositive but were never hospitalized for the treatment for a febrile illness (non-severe dengue, NSD=36) and in those with an episode of DHF in the past (severe dengue, SD=34). We also assessed the NS1 antibodies in dengue seronegative individuals (n=20). We found that individuals with past SD had significantly higher ($p=0.004$) NS1 antibody levels (median 1.6, IQR 0.78 to 2.2) when compared to those with NSD (median 0.91, IQR 0.65 to 1.36) (Fig 1a in the revised manuscript). However, we have no data regarding the infecting DENV serotype of these individuals and the number of past infections they had, which would probably influence the NS1 antibody levels.

The figure 5 described in this manuscript (now supplementary figure 6), were of healthy individuals living in the community and not of patients. As shown in supplementary figure 6, there were marked differences in the regions of NS1 recognized by healthy seropositive individuals who had inapparent (asymptomatic or mild dengue) and those who had a past history of DHF.

In the patient cohort although we have examined the NS1 antibody epitopes at the time of the critical phase, we agree with the reviewer that it would also be helpful to examine the NS1 antibody epitopes at the time of recruitment, before any of the patients had developed vascular leak (day 3 of illness). We found these new data interesting and are grateful for the suggestion. First of all, the NS1 antibody epitopes recognized by those who progressed to develop DHF were again very different from those with DF and these regions differed in patients with acute DENV1 compared to those with acute DENV2. In acute DENV1 infection, antibody responses to only 5 regions of DENV1 NS1 were different in those who progressed to develop DHF, compared to those with DF and only antibody responses to peptide 26 (aa 144-160) and peptide 61 (aa 341-353) were common in pre-DHF and at the time of vascular leak in those with DHF. In contrast to those with acute DENV1 infection, we saw marked differences in those with acute DENV2 on day 3 of illness, before any of the patients

had progressed to develop DHF. Patients who progressed to develop DHF, had significantly lower antibody responses to many regions of NS1, including the N terminal region and the C terminal region (new figure 5). These differences were more marked in peptide 33 and 34 (aa 236 to 251 and aa 242 to 257) and for peptide 12 in the N terminal region (aa 85-102). It is interesting that healthy seropositives in the community had with NSD had significantly lower antibody responses to the N terminal peptides of NS1 compared to healthy individuals who had an episode of DHF. However, these differences could be attributed to the differences in DENV serotypes they were exposed to or because those with inapparent infection in the community had a past primary infection, whilst those with past DHF were more likely to have had a secondary dengue infection. These differences can only be dissected by long-term large-scale prospective studies carried out in the community, capturing seroconversion and exposure events.

2. There are many risk factors involved in development of DHF. One of such factors was addressed by the author, secondary dengue virus infection based upon the IgM:IgG ratio. This is the major flaw of the definition. As of today, there has been reported that some of infected subjects, the IgM levels can be lasted for quite sometimes, that some of these subjects may not have response to the infection, and that some protected individuals are low in IgG levels. As such, the authors may need to perform the T cells, especially the memory T cells, to sort out the issue.

Response: We completely agree with the reviewer. We have used the WHO 2011 guidelines definition of secondary dengue infection as the IgM:IgG ratio as <1.2 . They further state that the cut-off value of <1.2 for differentiating primary and secondary dengue varies whether the patient has classical or non-classical dengue and that more recent data show that a ratio of <2.6 of IgM: IgG was 100% accurate in classification of classical dengue infection and 90% accurate in classification of non-classical dengue infections. All the patients whom we classified as having a secondary dengue infection and therefore, included in this study had IgM: IgG ratios <1.2 (and therefore <2.6). The recommendations of the WHO have been based on the findings of Falconar et al (Falconar et al, Clin Vac Imm 2006) and these investigators have shown that classification of patients as primary and secondary dengue based on IgM and IgG ELISA titres are more accurate than using ELISA optic density values. In our classification, we have used IgM and IgG titres of the ELISAs and not OD

values. All the patients included in this study were viraemic at the time of recruitment and therefore, they were having an acute dengue virus infection and are unlikely to have had IgM present from a previous infection. Serial NS1 antigen levels were also carried out in this cohort of patients using the Panbio NS1 capture ELISA and these data too have now been included (supplementary figure 1).

However, as the reviewer had requested to determine the association of T cell responses with relation to NS1 antibodies we have carried out these assays. As there were no PBMCs available from this cohort of patients, we recruited 23 new patients with acute secondary DENV1 infection (DF=11 and DHF=12) and 16 patients with acute secondary DENV2 infection (DF=4, DHF=12). We assessed their NS1 antibody responses to the recombinant DENV1 NS1 or the DENV2 NS1 protein based on the infecting serotype. IFN γ ex vivo ELISpots were carried out to determine T cell responses to DENV NS3, NS5 and NS1 overlapping peptides. NS3 and NS5 were chosen as they have been shown as the most immunodominant proteins and NS1 was chosen to find any association with NS1 specific T cell responses and NS1 antibodies. As described by many groups and us, DENV specific T cell responses were absent or present in a very low frequency in the majority of patients with acute dengue infection which associates with lymphopenia (Mongkolsapaya J, Journal of Immunology, 2006; Mongkolsapaya J, Nature Medicine, 2003; Malavige et al, PLOS Neg Trop Dis, 2013). Therefore, it would be unsuitable to use T cell responses in patients with acute dengue to define if the patients have a primary or secondary dengue infection, as it has been shown in all the above quoted studies and in many more that DENV specific T cells are present at very low frequency in acute infection, even in those with secondary dengue infection. There was no association with NS1 antibody levels with any NS3, NS5 or NS1 specific T cell responses in either acute DENV1 or DENV2 infection.

As the reviewer had specifically asked for the association of memory T cell responses with NS1 antibodies, we assessed the association of DENV specific T cell responses in a cohort of healthy DENV seropositive individuals. In a previous study, we had characterized the T cell responses in a community cohort of individuals who either had inapparent dengue infection (NSD = they were DENV seropositive, without ever being hospitalized for a febrile infection) or had DHF in the past

(SD) (Jeebandara et al, PLOS Neg Trop Dis, 2015). In this cohort of patients, we had only evaluated the T cell responses to DENV NS3 overlapping peptides as NS3 is considered the most immunodominant protein. However, we do not have any information regarding the past infecting serotype in these individuals or the number of past infections they had. As mentioned previously (Fig 1a in revised manuscript), those with past SD had significantly higher NS1 antibody responses than those with past NSD. We did not find any correlation of NS1 antibody responses with DENV-specific T cell responses. Although we had previously developed a T cell-based assay for determining the past infecting DENV serotype and therefore, to find out how many past infections an individual had been exposed to, we found that this assay was not suitable to be used in patients with acute dengue (Malavige et al, Clinical and Exp Immunology, 2012). Therefore, currently there are no T cell-based assays that can be used in acute dengue infection to differentiate between primary and secondary dengue infection. Although we have now generated new data on T cell responses in acute dengue and relationship with NS1 antibodies and similarly presented data on memory T cell responses in healthy individuals with varying severity of past infection and NS1 antibody levels, we believe these data should be treated with caution because of the low circulating T cell frequency.

Although we have carried out the above T cell-based experiments as requested by the reviewer, we have not included these data in the revised version, as we felt it could risk confusing the picture in the setting of lymphopenia, and it was difficult to justify the reasons for carrying out T cell-based assays to differentiate primary and secondary dengue.

Minor points:

1. The title of the manuscript is too broad and does not fit the contents in the manuscript.

Response: We wish to thank the reviewer for this comment. We have changed the title to ‘Role of NS1 antibodies in the pathogenesis of acute secondary dengue infections’ as suggested by the second reviewer.

2. Page 6, a reference should be added after the statement. “In DENV1-infected DHF patients the serum concentration of NS1 antibodies started to rise 5.5 days after the onset of illness.” This sentence is misleading since the onset of illness means fever, which has 5.6 days of

incubation time already. It is important to know that the recall antibody response is earlier in secondary dengue virus infected subjects. As such, the sentence required to be rephrased to reflect the actual time line since the time of virus inoculation.

Response: We thank the reviewer for this comment. We have changed the phrase from onset to illness to onset of fever throughout the manuscript.

3. The levels of dengue NS1 protein in circulation of the acute patient have been reported to be correlated with the disease severity. In current manuscript, the authors did not report the concentration of NS1 protein in these recruited patients. Importantly, the NS1 proteins in dengue patients has been shown to be hexamer formation, did authors check the NS1 antibodies bind to the NS1 harvested from acute patients? The critical questions will be “will the higher the concentration of NS1 protein result in higher levels of NS1 antibodies, and subsequent lead to severe dengue?” In addition, the viral load in these recruited patients have to be measured and presented.

Response: We thank the reviewer for this important comment. We too have reported that levels of NS1 protein in the circulation in patients correlated with disease severity (Paranavitane et al, BMC Infectious Diseases 2014). However, other studies have shown that this might not always be the case and that NS1 antigen levels are higher in patients with DENV1 infection and in patients with DF compared to those with DHF (Duong et al, PLOS Neg Trop Dis, 2011). Some studies have shown that the NS1 antigenaemia and viral loads are higher in DENV1 compared to DENV2 infections and persist for a longer time (Fox et al, PLOS Neg Trop Dis, 2011; Tricou et al, PLOS Neg Trop Dis, 2011). This study shows that NS1 antigen levels or viral loads were not higher in those with DHF. Although we had shown that NS1 antigen levels were likely to be higher in those with more severe forms of dengue, all the patients included in those studies had DENV1. Our recent data in DENV2 infection have shown varying results. In this cohort of patients, we found that NS1 antigen levels were lower in patients with DENV1 infection compared to DENV2 and in patients with DF in DENV1 infection the NS1 antigen levels persisted longer. In patients with DENV2 infection, the patients with DHF had higher NS1 antigen levels, which persisted longer. Similar results were seen for the dengue viral loads. The association of NS1 antigen with NS1 antibody levels in these patients have now been included in the revised version of the manuscript (supplementary figure 1).

have described in this study, were directed against the regions that were highly conserved among DENV1,2,3 and 4, JEV and WNV (Supplementary Fig 8a). The regions were similarly conserved when comparing the WNV, JEV and NS1 sequences of DENV1 and DENV2 strains that recently circulated in Sri Lanka (Supplementary Fig 8b). Therefore, the majority of NS1 antibodies defined in this study are likely to be highly cross-reactive. We did not see any difference in the antibody titres or preferential recognition of non-conserved or conserved regions in patients with either DF or DHF in acute disease. We have included these data in the results and discussion in the revised version of the manuscript.

5. In methods, the authors addressed that “Only those whose duration from onset of illness was ≤ 4 days were recruited.” But in the presented figure, all were shown more than 5 days of illness. Please clarified.

Response: We apologise if we caused some confusion. As shown in figure 1, all patients recruited with an acute secondary dengue infection presented on day 4 of illness (day 4 of fever) and all those with an acute secondary DENV2 infection presented on day 3 of illness (day 3 of fever) and these days have been indicated in the x-axis of the graphs. The patients with DENV2 infection typically presented to hospitals slightly earlier as during the epidemic it was observed that those with DENV2 were more likely to go into the critical phase earlier and the government gave health warnings to present to the hospital more quickly.

Reviewer #2 (Remarks to the Author):

In their manuscript, Jayathilaka et al. addressed the role of anti-NS1 antibodies in the pathogenesis of secondary dengue infections. Authors studied the anti-NS1 antibody kinetics in DF and DHF patients with secondary DENV-1 and DENV-2 infections and explored the anti-NS1 repertoires. Their results suggest that the epitope targeted by the antibody response could play an important role in the dengue pathogenesis and could lead to better vaccine and treatment design. These results are partially novel and of high interest in the field of dengue pathogenesis. The paper is clear, well-written and globally scientifically sound. Nevertheless, I do have some concerns that I would like the authors to address.

Postal Address: Post Office Box No. 6, Nugegoda, Sri Lanka.
Telephone: +94 11 2802026; Mobile: +94 77 2443193 Fax: + 94 11 2801480; Email: cdr@sjp.ac.lk

Major comments:

- People with past-asymptomatic dengue infection is one of the group studied here and I have some concerns with that. It is already very complex in a prospective study, with close monitoring, to identify dengue-infected people who remain asymptomatic, and I don't really understand how it is possible to ensure that people had an asymptomatic form of dengue using the health care facility as described in the M&M section? If the dengue episode was pauci-symptomatic or mild, most people may simply not recall the event after few weeks or months at the time they are tested at the health facility. It is therefore most likely that the group of individuals named "asymptomatic" or "inapparent" is a mix of people who experienced truly asymptomatic, pauci-symptomatic and mild disease. From a pathophysiological perspective, being asymptomatic and developing a mild form of the disease (most frequent form during primary infection in children) is not really the same. I would like the authors to clarify that point and since it may not be possible to include a real "asymptomatic" group to discuss how this may impact globally their conclusions.

Response: The reviewer highlights a very important point. At the time of recruitment for the study, some individuals had diagnosis cards of having DHF and their history was very clear. A large proportion of patients were found to be seropositive for dengue but had never been hospitalized due to a febrile illness. Although they would have had many episodes of febrile illness in the past, of which some of them may well have been due to a dengue infection, we agree with the reviewer that it would be incorrect to classify them as having asymptomatic dengue. However, as they did not have a severe dengue infection, we have now classified them as having non-severe dengue in the revised version.

- Results should be compared to those generated by the study of Hertz T et al., published in J Immunol (available online on April 2017) as these authors previously looked at the role of anti-NS1 antibodies with sometimes a similar approach. Interpreting the results of this study in the light of what Hertz already described would provide a better understanding of the complex mechanisms involved in dengue pathogenesis.

Response: Thank you for directing us towards this paper. As mentioned by the reviewer, these investigators have used a similar approach to us and examined the NS1 antibody epitopes of 10 children with DF (7 with primary dengue and 3 with secondary dengue) and 7 children with DHF (2 with primary dengue and 5 with secondary dengue). Similar to our observations they have found differences in the NS1 antibody epitopes in those with DENV1, DENV2 and DENV3 infection. Except for recognition of region 115 to 137 in our patients with an acute secondary DENV1 infection, none of the other regions identified by this group were similar to the main antibody binding regions identified by us, in acute dengue DENV1 or DENV2 infection. These differences could be attributed to us limiting our patient cohort to those who had a secondary dengue infection and also due to factors such as the sample size (their total sample size 26 vs our sample size of 42 used to identify NS1 antibody epitopes) and differences in the ethnicities and ages. We have included this in the discussion of the revised version of the manuscript.

- Authors should provide a little bit more alternative explanation regarding the differences in NS1 antibody kinetic between DENV-1 and DENV-2, e.g. the serotypes involved in previous infections/sequence of previous infections; the variations between different strains of the same serotype on NS1 production (Watanabe, J Virol 2012), etc.

Response: This is a very important comment. We have done multiple alignment of the DENV1 and DENV2 sequences and while some of the antibody epitopes are similar to both viruses, some antibody epitopes which were only found to be specific to either DENV1 or DENV2, were directed against regions that were non-conserved. We have added this to the results and the discussion in the revised version of the manuscript and have included a supplementary figure 3 of these alignments.

Minor comments:

- Title should be “Role of NS1 antibodies in the pathogenesis of acute secondary dengue infections” (no primary cases were included in the study)

Response: Thank you for this comment. We have changed the title as specified by the reviewer.

- Please be consistent with the use of DENV rather than DEN when referring to the dengue virus

Response: We apologise for the inconsistency. We have changed this in the revised version.

- Line 47: this sentence should start with “In addition” rather than “Indeed” as the increased likelihood of disease enhancement is not related to the 2 issues mentioned in the previous sentence (poor efficacy in naïve patients and efficacy depending on serotype)

Response: Thank you for this comment. We have changed the wording as advised.

- Line 49: NS1 is also a marker of dengue infection (can be found in asymptotically-infected individuals), not only of disease.

Response: Thank you for this valid comment. We have changed it in the revised version as recommended.

- Line 191: “p=” is missing for the peptide pools 2

Response: Thank you for pointing this out. We have corrected this mistake.

- Line 312: What about impact on platelet counts observed in DF patients with DENV-2 infection?

Response: We apologise if this is not clear. The association of NS1 antibodies in platelets in DF patients with DENV2 is described just after the description about them in DHF as Spearman’s $r=0.3$, $p=0.03$ (revised manuscript line 177-179).

- Discuss briefly the result limitations since only DENV-1 and DENV-2 infections were studied.

Response: Thank you for this comment. We have included this in the discussion.

- For the in-house assays developed for the purposes of this study, more details on how the threshold were set (positive and negative controls), the wavelengths used by the reader, if

samples were tested in duplicates, triplicates, etc. would help the reader to develop the assays but also to better trust the results.

Response: We apologise for not providing more details, which we have now done in the revised version of the manuscript.

REVIEWERS' COMMENTS:

Reviewer #1 (Remarks to the Author):

All concerns raised had been clarified and addressed.

Reviewer #2 (Remarks to the Author):

Thank you for addressing my comments. I'm satisfied with the revised version of the manuscript.